# Ultra-low-current driven InGaN blue micro light-emitting diodes for electrically efficient and self-heating relaxed microdisplay

Woo Jin Baek [1], Juhyuk Park[1], Joonsup Shim[1], Bong Ho Kim[1], Seongchong Park[2], Hyun Soo Kim[1], Dae-Myeong Geum [3,4,5] ✉ & Sang Hyeon Kim [1,3,5] ✉

InGaN-based micro-light-emitting diodes have a strong potential as a crucial building block for next-generation displays. However, small-size pixels suffer from efficiency degradations, which increase the power consumption of the display. We demonstrate strategies for epitaxial structure engineering carefully considering the quantum barrier layer and electron blocking layer to alleviate efficiency degradations in low current injection regime by reducing the lateral diffusion of injected carriers via reducing the tunneling rate of electrons through the barrier layer and balanced carrier injection. As a result, the fabricated micro-light-emitting diodes show a high external quantum efficiency of 3.00% at 0.1 A/cm² for the pixel size of $10 \times 10 \ \mu m^2$ and a negligible $J_{max \ EQE}$ shift during size reduction, which is challenging due to the non-radiative recombination at the sidewall. Furthermore, we verify that our epitaxy strategies can result in the relaxation of self-heating of the micro-light-emitting diodes, where the average pixel temperature was effectively reduced.

Recently, as the demand for ultrahigh-resolution display in near-eye applications such as augmented reality/virtual reality (AR/VR) has highly increased, group-III nitride (N)-based micro light-emitting-diodes (μLEDs) have received much attention as one of the strong candidates for the self-emissive light source, due to their outstanding performances of high resolution, high luminous efficiency and brightness, fast response speed, and robust durability[1–5]. Among micro-displays, for realizing these near-eye displays which require above 2000 pixel-per-inch (PPI), pixel scaling of μLEDs to ultra-small size (<10 μm) is inevitable. However, unlike LEDs for conventional lighting applications, critical problems have emerged with the size shrinkage to a few micrometers which means a high surface-to-volume ratio. Specifically, efficiency degradation phenomena such as the decrease of maximum external quantum efficiency (EQE) and the shift of current density showing maximum EQE ($J_{max \ EQE}$) to higher current density regions was reported in several studies when the device size of

μLEDs decreased[6–8]. These problems are fatal to the implementation of high-resolution displays since they could induce high power consumption and self-heating during the display operation. Indeed, the power consumption and heat generation are not distinct issues since the generated heat could cause an increase of a junction temperature, emission wavelength shift, and decrease of EQE of LEDs.

Thus, to circumvent the above issues, many kinds of research have been reported to suppress the side effect with a reduction in the size of μLED. Wong et al. showed the surface passivation method to alleviate the damage caused by the plasma-assisted dry etching process[7]. Also, Park et al. introduced a pixelization method using heavy ion implantation to avoid plasma damage at the sidewall[9]. From these results, although there were improvements regarding peak EQE values and reduced dark current, the proposed methods must rely on post-growth techniques, where the perfect sidewall passivation method or the defect-free pixelization remains to be questionable.

[1]School of Electrical Engineering, Korea Advanced Institute of Science and Technology (KAIST), Daejeon 34141, Republic of Korea. [2]Division of Physical Metrology, Korea Research Institute of Standards and Science, Daejeon 34113, Republic of Korea. [3]Information and Electronics Research Institute, Korea Advanced Institute of Science and Technology (KAIST), Daejeon 34141, Republic of Korea. [4]School of Electronics Engineering, Chungbuk National University, Chungcheongbuhk-do 28644, Republic of Korea. [5]These authors jointly supervised this work: Dae-Myeong Geum, Sang Hyeon Kim.
✉e-mail: dmgeum@chungbuk.ac.kr; shkim.ee@kaist.ac.kr

Therefore, to fundamentally solve the size-dependent efficiency degradation, it is necessary to ultimately reduce the surface effect by conceiving new epitaxial structures for µLEDs based on the carrier transport behavior in small-size devices. In the meanwhile, most previous studies on the design and optimization of III-N epitaxial structure were motivated by high-power lighting applications for the last two decades, which focused on large chip size, high power, and high operating current region. Various epitaxial structure engineering methods including modification in quantum wells (QW)[10,11], quantum barriers (QB)[12,13], electron blocking layers (EBL)[14–16], and growth substrates[17–19] have been proposed to mitigate the efficiency droop in the high injection current density region. Although the proposed structures improved the efficiency droop and peak EQE values, they may not be equivalently adopted in the use of µLED applications due to the difference in optimal operating current density regions. For these reasons, a reconsideration of epitaxial structures for µLEDs display must be carried out before post-growth optimization, but it has not been seriously considered.

Thus, in this paper, we propose fundamental epitaxy strategies to effectively suppress size-dependent issues such as EQE reduction and current density shift showing peak EQE in µLEDs, by newly developing epitaxial structures including QB and EBL. First, from the point of view of carrier confinement, we systematically investigated the effect of QB thicknesses on the maximum EQE and $J_{max\ EQE}$ by varying the QB thicknesses. The experimental results indicated that thick QB thickness can effectively manipulate the QB barrier height and tunneling rate for increasing carrier confinement so that the EQE values and $J_{max\ EQE}$ shift are noticeably improved. Furthermore, based on the composition engineering in the EBL layer, a balanced injection of carriers was achieved by reducing the polarization charge and lowering the hole barrier. By utilizing these optimal structures, even without the sidewall passivation, the plot of size-dependent EQEs showed a negligible $J_{max\ EQE}$ shift indicating a highly suppressed sidewall effect even in $10 \times 10\ \mu m^2$ devices with the high EQE in low current density $(0.1\ A/cm^2)$. Finally, we investigated the thermal properties of the µLEDs to see the impact of the EQE on self-heating. Through this epitaxial structure engineering, we successfully reduced the heat generation by a low driving current which would allow high reliability and durability. Based on these findings, we believe that the proposed strategies to reduce surface effects will be an ambitious approach for the future ultra-low current-driven micro-displays.

## Results and discussion

### Design of the quantum barrier

QB plays an important role in the active region of LEDs as a carrier confinement layer. Many efforts such as thickness modification, polarization balancing, and doping concentration have been made to solve the efficiency droop of GaN-based LEDs in high current density regions. Several studies have reported that decreasing QB thickness mitigates the efficiency droop in large-sized LEDs because they not only enhance the overlap of electron and hole wavefunctions but also decrease the kinetic energy which assists the escape of carriers induced by the internal electric field in thinner QBs[20–22]. In fact, besides the wavefunction and the kinetic energy, QB thickness scaling would also lead to the formation of the miniband, change in barrier height in the QW energy band and the tunneling rate, which must be taken into account in the epitaxial structure design in µLEDs. The tunneling rate ($R_T$) of a carrier between the QWs can be described by the zero-order Wentzel-Kramers-Brillouin (WKB) approximation[21,23]:

$$R_T = \frac{\exp(-\int_0^{L_b} 2\hbar^{-1}\sqrt{2m^*\triangle u}\,dx)}{L_w}\sqrt{2E_0/m^*} \qquad (1)$$

where $L_b$ is the QB thickness, $\hbar$ is the reduced Planck's constant, $m^*$ is the effective carrier mass, $\Delta u$ is the QB energy height, $L_w$ is the QW thickness, and $E_0$ is the ground energy state of QW. When the QB thickness increases, the energy barrier also increases due to the conservation of the electrostatic field (see Supplementary Fig. 1). Additionally, based on Eq. (1), the tunneling rate of a carrier decreases with increasing QB thickness and barrier height which indicates the probability of electrons being confined in the QW increases with increased QB thickness. Thus, the correlation between internal field distribution and tunneling rate induced by QB thickness variation must be considered.

To investigate the practical effect of QB thickness on characteristics of µLEDs, we designed epitaxial structures with different QB thicknesses of 3.5 nm, 7.5 nm, and 10.5 nm, (namely QB 3.5, QB 7.5, and QB 10.5), while remaining other layers completely identical as shown in Fig. 1a. Figure 1b shows the cross-sectional transmission electron microscopy (TEM) images of the epitaxially grown full LED structure of QB 3.5 sample. It was clearly confirmed that the epitaxial layers are uniformly grown with the target design thickness of each layer. Figure 1c–e shows the magnified TEM images of the active region of samples with different QB thicknesses (The magnified version of TEM images can be found in Supplementary Fig. 2). The normalized energy dispersive spectroscopy (EDS) intensities for indium atoms are shown in Fig. 1c–e along the red lines, suggesting the formation of abrupt interfaces between QBs and QWs without material intermixing in all three different samples. Furthermore, we investigated the optical properties of grown epitaxial structures. Figure 1f shows photoluminescence (PL) spectra of each structure with an incidence of 325 nm He-Cd laser at room temperature. The peak wavelengths of 423.5 nm, 440.7 nm, and 444.8 nm were observed in QB 3.5, QB 7.5, and QB 10.5, respectively. With a reduction of QB thickness, the blue shift of the peak PL wavelength was observed due to the smaller polarization effect in QW[24]. In addition, it is worth noting that the decrease of PL intensity and strong blue shift of peak wavelength were observed in the scaled QB thickness of 3.5 nm, indicating relatively low radiative recombination compared to QB 7.5 and QB 10.5 samples. This is because of the formation of miniband due to the coupling of wavefunctions between adjacent QWs. The formation of miniband can result in the strong blueshift of PL spectra and the reduction in radiative recombination[25]. From this result, we confirmed that thick QB would lead to improved radiative recombination with better carrier confinement in QW, although there were some studies suggesting thin QB has better confinement due to lower internal field[21].

To evaluate the effect of the carrier confinement in QB thicknesses on actual µLEDs, we fabricated the scaled µLEDs with a lateral structure using three different epitaxial structures as shown in Fig. 1g and Fig. 1i. With conventional photolithography and Cl$_2$-based dry etching, the successful fabrication of the scaled µLEDs with a size down to $10 \times 10\ \mu m^2$ is carried out. Figure 1h shows the top-view optical microscopic image of $10 \times 10\ \mu m^2$ µLEDs for QB 10.5 without and with current injection. The clear emission of blue light was observed in the fabricated µLED, indicating the small size µLED can be successfully demonstrated using the fabrication process even without sidewall passivation.

Figure 2a shows the logarithmic current density versus voltage (J-V) characteristics of QB 3.5, QB 7.5, and QB 10.5 with the size of $80 \times 80\ \mu m^2$ and $10 \times 10\ \mu m^2$. The dark current in the reverse bias region was measured to the measurement noise floor and showed a low dark current, which implies no significant leakage path in the fabricated µLEDs. The thinner QB samples showed a higher current density when the forward bias is over 3 V, and µLEDs with the size of $10 \times 10\ \mu m^2$ showed a higher current density than $80 \times 80\ \mu m^2$ sized devices. We observed a small but gradual increase of current density at a voltage approximately ranging from 2.2 V to 3 V in decreasing the size of the µLEDs (see Supplementary Fig. 3). We noticed the linearity of

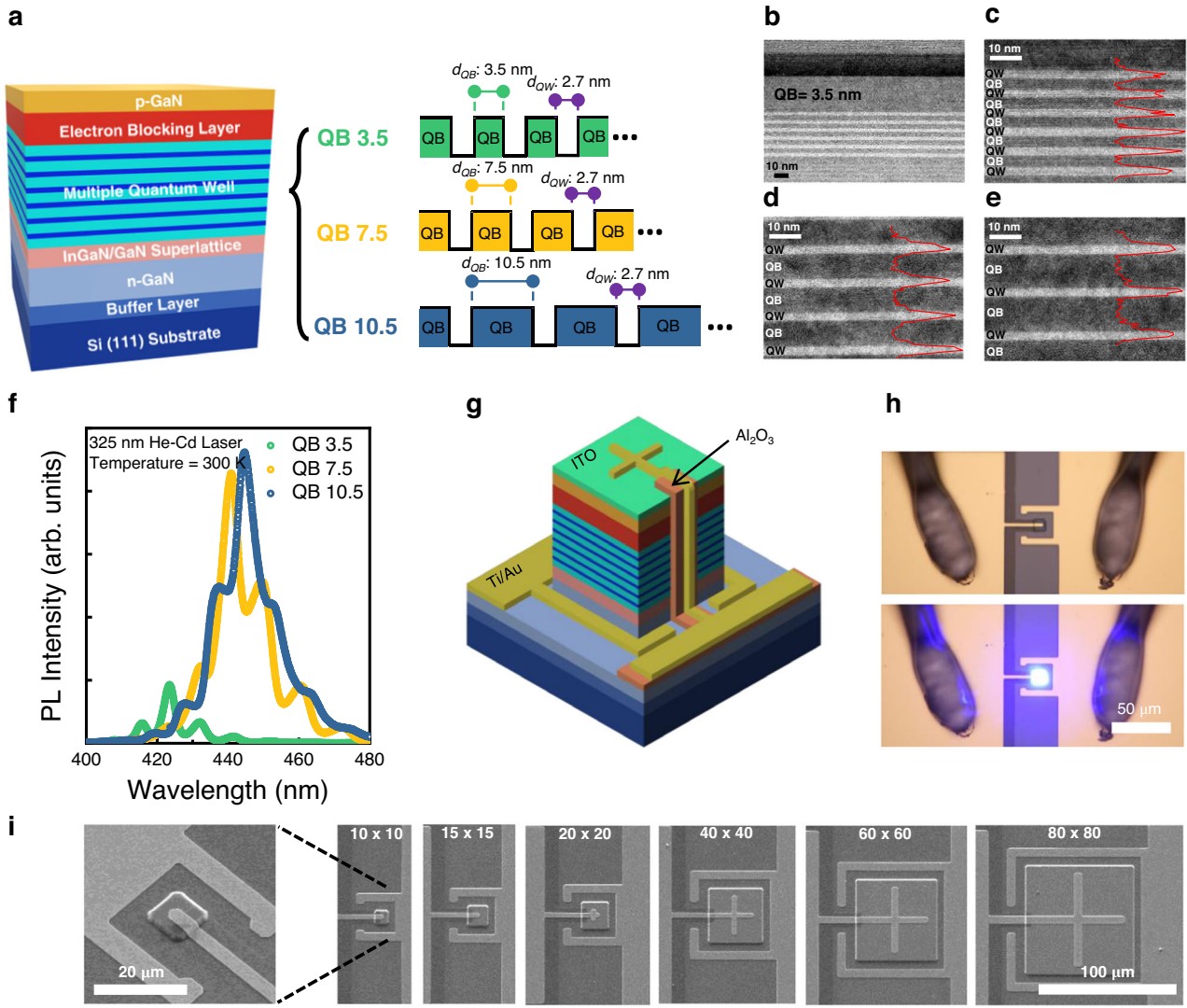

**Fig. 1 | Characterization of grown µLED epitaxy for the redesign of quantum well barrier in µLED. a** Schematic of epitaxy structure and designated active region structure with different QB thicknesses. **b–e** Transmission electron microscope (TEM) images of grown epitaxy layers (**b** from superlattice layers to p-GaN of QB 3.5. **c–e** Active region of **c** QB 3.5 **d** QB 7.5 **e** QB 10.5). The red line in **c–e** shows the line-scanned energy disperse spectroscopy of the indium component. **f** Photoluminescence (PL) spectra of QB 3.5, QB 7.5, and QB 10.5. **g** Schematic of fabricated devices. **h** Top−view optical microscope image of fabricated 10 × 10 µm² µLED devices without and with injection current. **i** Scanning electron microscope (SEM) images of the fabricated µLEDs in different sizes. (the scale bar in 80 × 80 image applies to all top-view SEM images in **i**).

size-dependent increase of current density, and was possible to investigate the components of forward current as shown in the inset of Fig. 2b. The forward current in a mesa-shaped diode is composed of bulk forward current per junction area ($J_B$) pathing through the mesa and surface current ($J_S$) pathing around the mesa. Therefore, the total forward current of a mesa-shaped diode can be expressed as Eq. (2)[26]:

$$I = J_B \times S + J_S \times L \qquad (2)$$

where $I$, $S$, and $L$ stands for total current, junction area, and perimeter of µLEDs mesa, respectively. From equation (2), $J_B$ and $J_S$ can be extracted by linear-fitting the slope of the $S/L$-$I/L$ and $L/S$-$I/S$ relationship (see Supplementary Fig. 4). Figure 2b shows the $J_S$ as a function of the applied bias $V$. As the QB thickness decreased, the surface current increased as a higher voltage is applied as shown in Fig. 2b. In order to identify the ratio of surface current to bulk current, the $J_S/J_B$-$V$ curve was plotted as shown in Fig. 2c showing a clear trend of $J_S/J_B$ of QB 3.5 higher than QB 7.5, and QB 7.5 higher than QB 10.5. When the applied voltage is higher than 2.45 V, the dominance of surface current

increases with increasing voltage as the QB thickness decreases, meaning injected carriers cannot participate in the radiative recombination process. Consistently, as shown in Fig. 2d, the maximum EQE of QB 3.5 was recorded at 4.24% and 2.90% for 80 × 80 µm² and 10 × 10 µm², while QB 7.5 recorded 4.74% and 3.01%, and QB 10.5 recorded 5.12% and 3.12% for 80 × 80 µm² and 10 × 10 µm², respectively. The fabricated µLED showed a lower EQE in the thinner QB samples compared to the thicker QB samples, which was attributed to the surface leakage suppression which was confirmed in our previous research for sidewall gating effect[27]. More importantly, it was observed that QB 3.5 had a higher $J_{max\ EQE}$ shift in smaller device sizes than these of QB 7.5 and QB 10.5 ($J_{max\ EQE}$ region is shadowed with green color). When the size reduced from 80 × 80 µm² to 10 × 10 µm², $J_{max\ EQE}$ shifted from 55.5 A/cm² to 250.0 A/cm² in the case of QB 3.5, while a smaller peak shift of from 15.6 A/cm² to 35.0 A/cm² and from 10.2 A/cm² to 30.0 A/cm² was observed in the case of QB 7.5 and QB 10.5, respectively. As shown in Fig. 2e, the results of maximum EQE and $J_{max\ EQE}$ indicate that noticeable improvements can be achieved depending on the QB thicknesses. Figure 2f depicts the

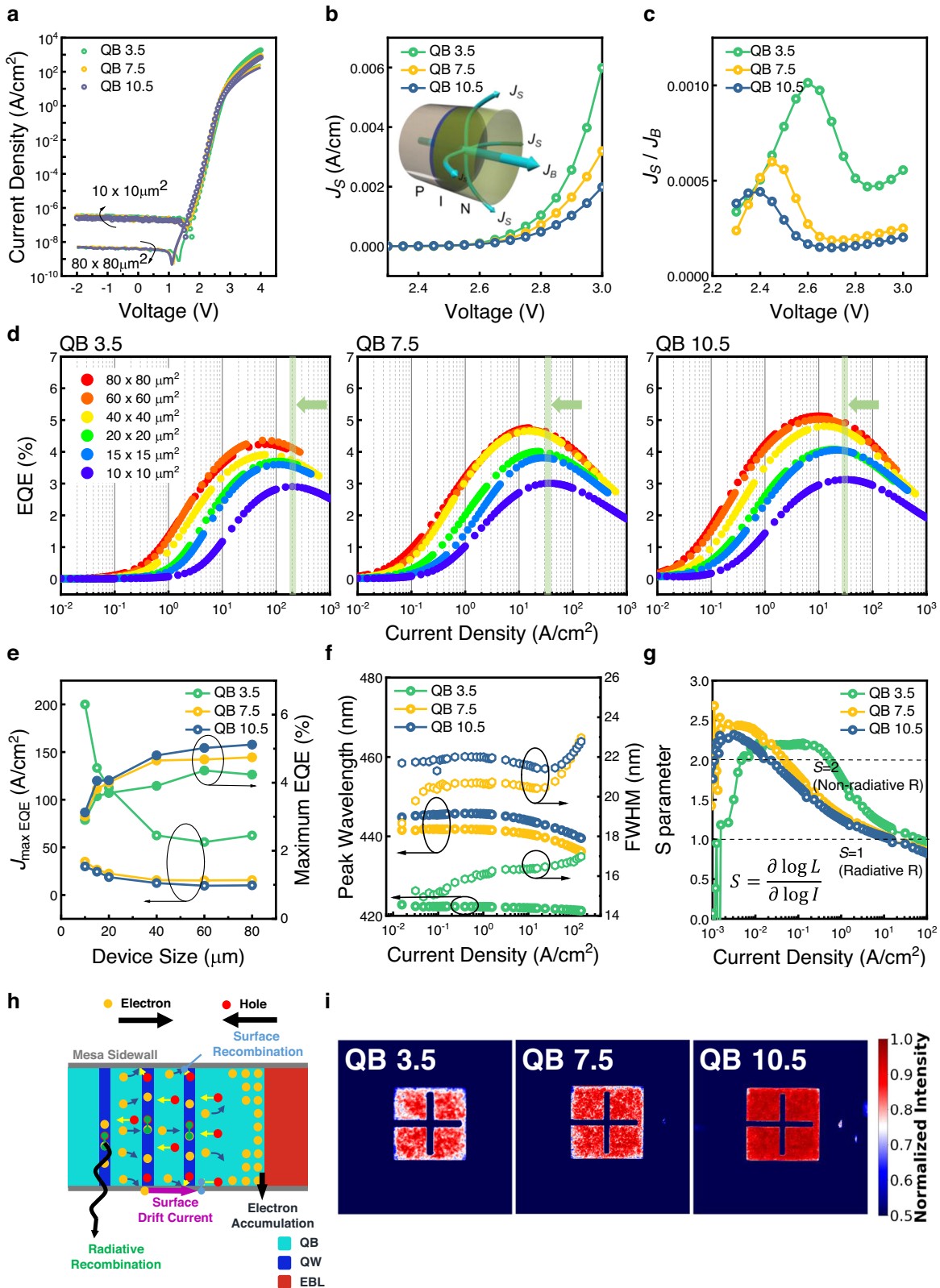

electroluminescence (EL) peak wavelength and the full width at half maximum (FWHM) value as a function of current density for devices with different QB thicknesses (See supplementary Fig. 5 for EL spectra at different current densities). A smaller blue shift of EL peak was observed in QB 3.5 than QB 7.5, and QB 10.5, which is due to the difference in polarization effect as explained in Fig. 1f. Additionally, all three devices showed evidence of the screening effect and band-filling

effect, but in varying degrees. Smaller screening effect and band-filling effect were observed in thinner QB thicknesses[21,22]. Despite this factor, the peak wavelength shift and change in FWHM for QB 10.5 in the low current density region were negligible, which means no chromaticity change in low current density.

To investigate the dominant recombination mechanism according to the injection current, we employed the $S$ parameter (defined as

**Fig. 2 | Electrical and optical characteristics of QB 3.5, QB 7.5, and QB 10.5.**
**a** Logarithmic *J-V* characteristic of µLEDs with a pitch size of $80 \times 80\ \mu m^2$ and
$10 \times 10\ \mu m^2$. **b** $J_S$ and **c** $J_S/J_B$ as a function of voltage for three different samples. (inset
of **b**. Schematic image of PIN LEDs and the components of forward current paths.)
**d** EQE-logarithmic current density curve for QB 3.5, QB 7.5, and QB 10.5 with pitch
sizes from $80 \times 80\ \mu m^2$ to $10 \times 10\ \mu m^2$. (The green line shows the point where
$10 \times 10\ \mu m^2$ device is at its maximum EQE) **e** $J_{max\ EQE}$ and maximum EQE of devices
with different sizes. **f** Electroluminescence peak wavelength and FWHM. **g** *S* para-
meter as a function of logarithmic current density. **h** Schematic diagram of possible
carrier transport processes in µLEDs under forward bias. The electrons drifting or
tunneling throughout the active region can diffuse to the sidewall where the sur-
face recombination can occur or drift through the surface leakage path. The drifted
or tunneled electrons through the active region can also accumulate at the inter-
face between the active region and the electron blocking layer, where a lateral
diffusion can be significant. **i** Normalized electroluminescence intensity mapping of
QB 3.5, QB 7.5, and QB 10.5. The pitch size of the device is $80 \times 80\ \mu m^2$, and the
operating current density is $0.1\ A/cm^2$.

$S = \partial log(L)/\partial log(I))$ in Fig. 2g, which describes the dominant carrier recombination mechanism[28]. Figure 2g shows the *S* parameter of QB 3.5, QB 7.5, and QB 10.5 obtained from $80 \times 80\ \mu m^2$ sized devices (see Supplementary Fig. 6 for more information). When the *S* parameter is larger than 2, it means that the tunneling or surface leakage current corresponding to the non-radiative recombination is dominant[29]. The *S* parameter of QB 3.5 exceeds 2 over a wider range of current density region and is reduced below $S = 2$ at only the higher current density region, implying that carriers are not effectively injected into the QWs to occur radiative recombination in the low current density region. In contrast, QB 7.5 and QB 10.5 shows a relatively rapid approach to *S* parameter of 1, which means a more efficient injection in the lower current density regime. These results would be also consistent with the previous surface leakage current characterization.

In order to explain the relationship between the QB thickness and the different size-dependent electrical behavior in scaled µLEDs based on our experimental data with different QB thicknesses, the possible carrier transport mechanism is illustrated as shown in Fig. 2h. The higher $J_S/J_B$ (Fig. 2c) and lower EQE (Fig. 2d) in µLEDs with thinner QBs can be attributed to the difference in the tunneling rate of thickness-controlled QBs. If the tunneling rate of carriers is high, the probability of carriers being radiatively recombined (emission of carriers) can be decreased, resulting in the bypass of the carriers to the surface. Then, due to the formation of an n-type-like conducting surface layer in the sidewall of µLEDs caused by plasma damage[30,31], the bypassed carriers can diffuse laterally to the sidewall due to the bent energy band in the sidewall, contributing to the surface current which drifts along the mesa sidewall. Therefore, with QB scaling, the lateral diffusion of carriers to the sidewall throughout the mesa can be higher and the surface current pathing around the mesa cannot be effectively blocked, resulting in an increase in surface recombination at the sidewall and a decrease in current injection efficiency. Eventually, the lateral diffusion of carriers and surface current can cause a significant reduction in EQE due to the increased non-radiative recombination throughout the mesa sidewall. Furthermore, the formation of miniband in QB 3.5 can accelerate the lateral diffusion of carriers, since the carriers can tunnel through the miniband without being recombined[25]. As a result, $J_{max\ EQE}$ shift to the high current density region was much higher when the device size decreased and the maximum EQE was smaller in µLEDs with thinner QBs as depicted in Fig. 2e. To be mentioned, enhancements in device performances were observed after the sidewall passivation, but the trend of $J_S/J_B$ and device performances of QB thickness-varied devices were still valid (see Supplementary Figs. 7–10 in the Supplementary Information), which means that the epitaxy engineering is not an option but a must for µLEDs display. Additionally, to visualize the effect of surface current in devices with different QB thicknesses, the electroluminescence (EL) intensity mapping was conducted as shown in Fig. 2i. From the figure, it is possible to observe a lower normalized EL intensity at the surfaces near the sidewall in QB 3.5. A similar finding was also observed in QB 7.5, but the low normalized EL intensity near the sidewall was less severe than in QB 3.5. In the case of QB 10.5, a uniform EL intensity was observed throughout the device region. From the experiment on QB thickness variation, it clearly suggests a severe size-dependent efficiency degradation happens with a decrease in QB thickness, while the saturation of device performances could occur with an increase of QB thickness. For further optimization of device performances, QW thickness is one of the important parameters. However, QW thickness variation is more complicated because it also provides a trade-off between several important physical properties such as tunneling rate, barrier height, energy states, overlaps of carrier wavefunctions and critical thickness.

As a result, the correlation between electrical characterizations of $J_S/J_B$, *S* parameter, and EQE shows that the effective injection of carriers in the low current density region is important for low-current driven µLEDs.

## Design of the electron blocking layer

While the modification of QB thickness enhances the confinement of electrons, an effective injection of holes should also be considered where EBL plays an important role. During past decades, the escape of electrons from the QWs was known as one of the major physical origins of the efficiency droop in LEDs[32]. Wide-band gap p-doped AlGaN EBLs are included in the GaN-based LED epitaxial layers to prevent the escape of electrons passing to the anode. However, as pointed out in several previous studies, due to the polarization mismatch between the last QB and the EBL, a strong positive sheet charge is caused, resulting in a downward bending of the conduction band and electron accumulation in the corresponding interface[16,33]. Simultaneously, this downward band bending increases the hole barrier, which hinders the injection of holes into the active layer[15]. Such ineffective EBLs can aggravate the injection balance of carriers which can cause low radiative recombination in QWs due to low hole density. On the other hand, the composition and doping concentrations in the EBL would be an important parameter because the ionization energy of p-type Mg dopant can differ in different Al ratios of AlGaN[34], and the polarization mismatch at the MQW/EBL interface highly depends on Al composition[15,32]. Thus, considering the above effects, the Al composition and doping concentration of EBL need to be re-designed for the balanced injection of electrons and holes in the low current regime. For the systematic study, we have changed Al composition of EBL and doping concentration in order to reduce the hole injection barrier at the interface. Additionally, we increased the thickness of EBL to compensate for the decreased bandgap of EBL due to lowered Al composition considering the leakage current. The newly designed epitaxial structure is identical to the QB 10.5 sample, except for the EBL structure, and have named the sample QB 10.5 Balanced EBL. The detailed EBL structures are described in Methods section. Figure 3a shows the cross-sectional TEM image of the grown epitaxy layer of QB 10.5 and QB 10.5 Balanced EBL, confirming the epitaxy is grown as designated. In order to verify the carrier distribution along the active region in the corresponding epitaxial structures, numerical calculation of the energy band diagram, electron and hole concentration, and radiative recombination rate was carried out by using Atlas TCAD simulation from SILVACO, Inc. Figure 3b–f shows the TCAD simulation results of QB 10.5 and QB 10.5 Balanced EBL at the same current density of $0.1\ A/cm^2$. Figure 3b shows the energy band diagram of the two different epitaxial structures. Due to the high indium composition and lower doping concentration in QB 10.5, an energy valley is formed at the interface between the spacer and EBL, which leads to a high electron accumulation and barrier for hole transport in the corresponding

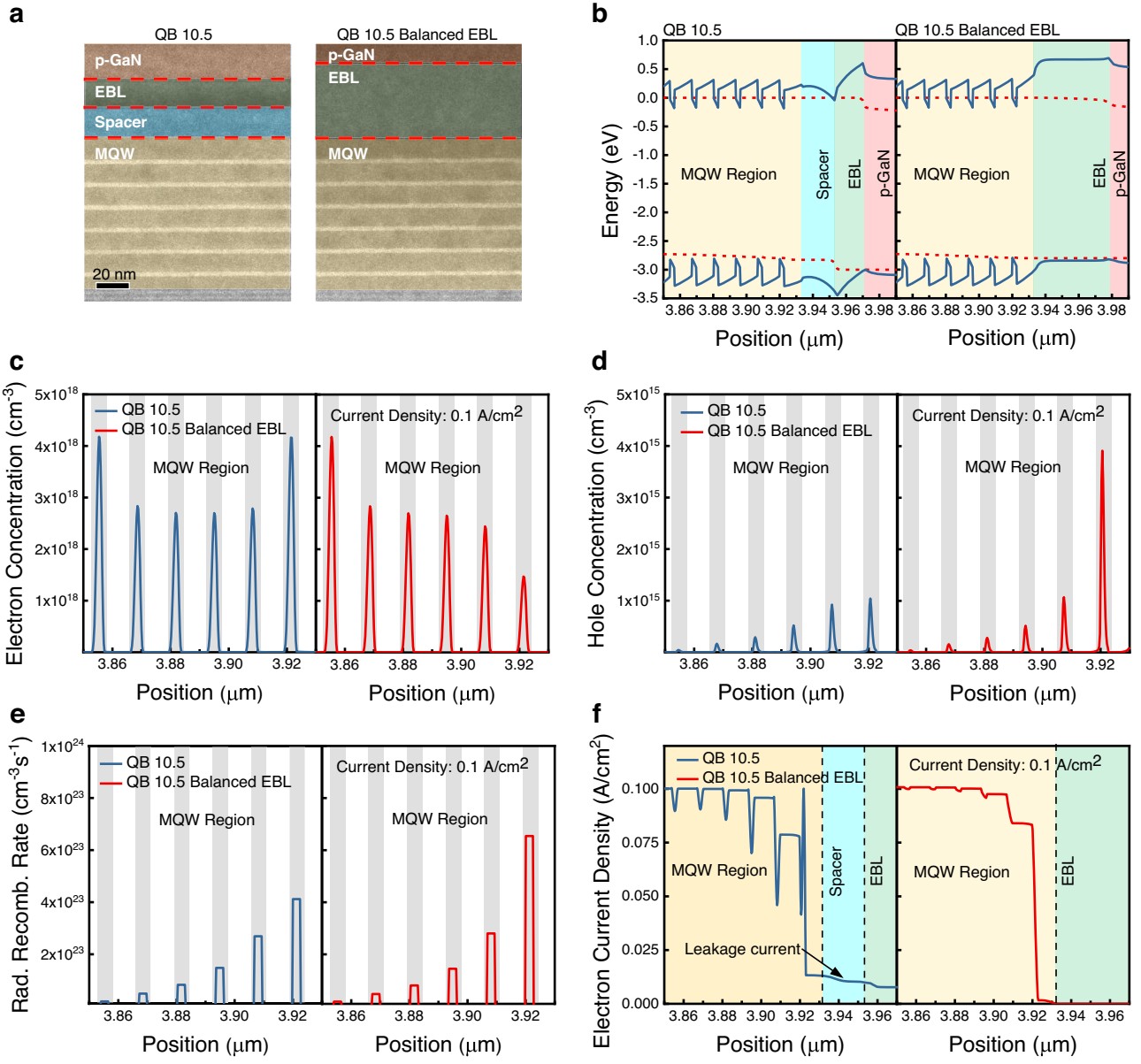

**Fig. 3 | Modified electron blocking layer. a** Transmission electron microscope (TEM) images of QB 10.5 and QB 10.5 Balanced EBL. **b–f** Simulation results of **b** energy band diagram **c** electron concentration **d** hole concentration **e** radiative recombination rate (the shaded part in **c–e** corresponds to the quantum well.) **f** electron current density of QB 10.5 and QB 10.5 Balanced EBL at the current density of 0.1 A/cm².

interface. As a result, while the electron concentration in the last two QWs in QB 10.5 Balanced EBL has decreased (Fig. 3c), due to the increased hole injection efficiency and hole concentration (Fig. 3d), the radiative recombination has increased compared to QB 10.5 (Fig. 3e), which implies that the balance of carrier population is more important in low current density region. Due to the increased radiative recombination and the reduced electron accumulation in the QB 10.5 Balanced EBL, the electron current density at the region after the MQW region (spacer and EBL) has dramatically decreased compared to QB 10.5 (Fig. 3f) (Detailed simulation results on the spacer layer can be found in Supplementary Fig. 11). The electron leakage here in the QB 10.5 can contribute to the surface current which can decrease the efficiency of the μLEDs as mentioned in the previous section.

Figure 4a shows the *J-V* characteristics of QB 10.5 and QB 10.5 Balanced EBL with the size of 80 × 80 μm² and 10 × 10 μm². By analyzing the *J-V* characteristics using the same method mentioned in the previous section, we found that the $J_S$ and $J_S/J_B$ of QB 10.5 Balanced EBL

at the forward bias smaller than 2.9 V (for $J_S$) and 2.95 V (for $J_S/J_B$) are lower than QB 10.5 as depicted in Fig. 4b, c, which indicates that the ratio of surface leakage current to the μLEDs mesa sidewall is much lower at small applied bias. However, since the bandgap of EBL has been decreased, the ability of EBL to block electron overflow may be debilitated. As a result, QB 10.5 Balanced EBL shows higher current density and $J_S/J_B$ when the forward bias is high. Nevertheless, the designed epitaxial structure exhibits low surface current in the low current density regime which is an advantageous property in epitaxial structure for display applications, and it was consistent in the optical properties. Figure 4d shows the EQE as a function of logarithmic current density. QB 10.5 Balanced EBL devices showed higher EQE than those of QB 10.5 for all sizes, and the $J_{max\ EQE}$ was much lower than QB 10.5. While $J_{max\ EQE}$ shifted from 10.2 A/cm² to 30.0 A/cm² when device size decreased from 80 × 80 μm² to 10 × 10 μm² in QB 10.5, a very small shift from 3.1 A/cm² to 4.0 A/cm² was observed in case of QB 10.5 Balanced EBL as shadowed in green color. The results emphasize that

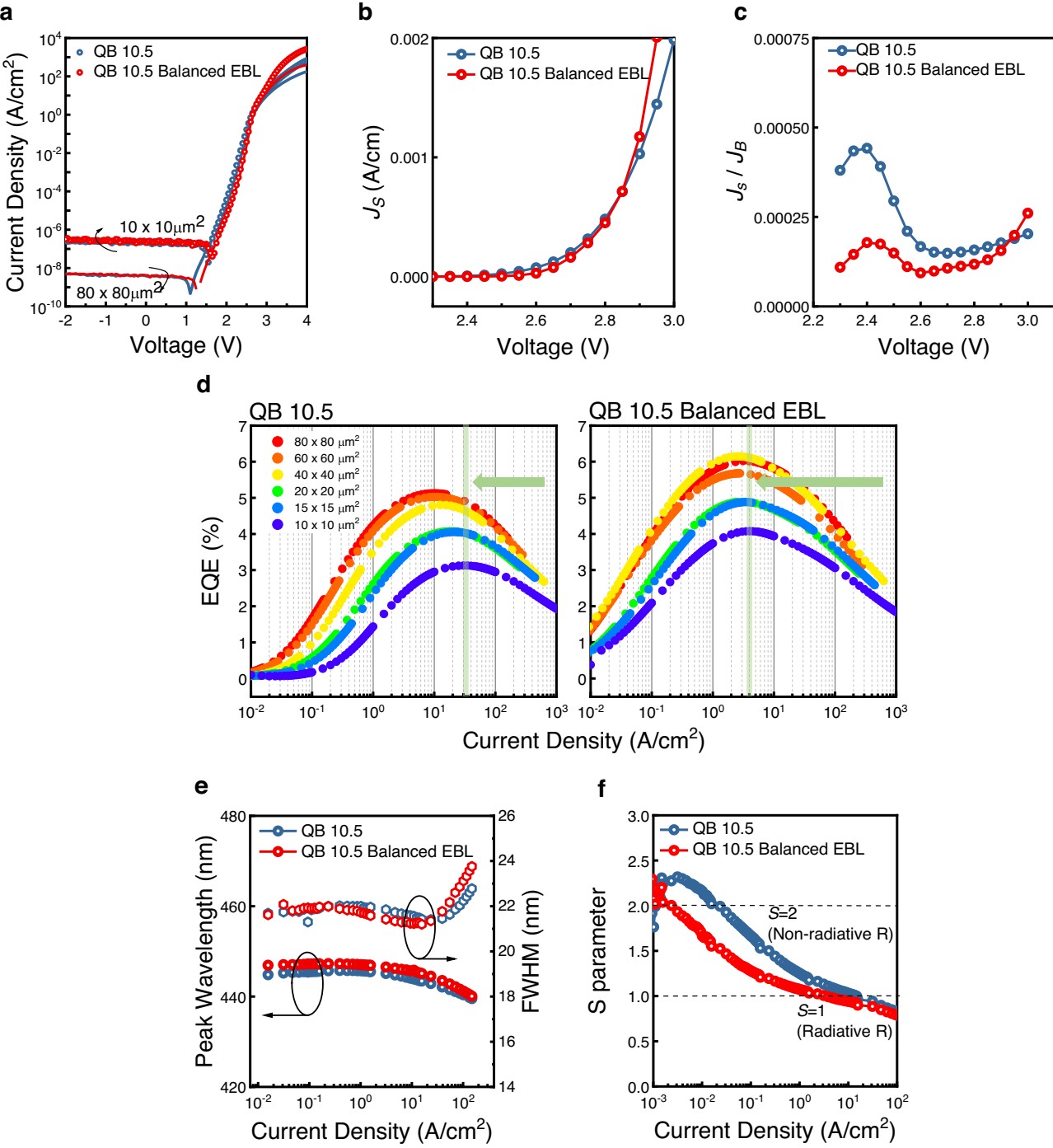

**Fig. 4 | Electrical and optical characteristics of QB 10.5 and QB 10.5 Balanced EBL. a** Logarithmic *J-V* characteristic of QB 10.5 and QB 10.5 Balanced EBL with pitch size of 80 × 80 μm² and 10 × 10 μm². (inset. normalized electroluminescence wavelength of μLEDs with a pitch size of 80 × 80 μm² driven at 1 mA.) **b** *Js* as a function of voltage **c** *Js/JB* as a function of voltage. **d** EQE-logarithmic current density curve for QB 10.5 and QB 10.5 Balanced EBL with pitch size from 80 × 80 μm² to 10 × 10 μm². (The green line shows the point where the 10 × 10 μm² device is at its maximum EQE) **e** electroluminescence peak wavelength and FWHM. **f** *S* parameter as a function of logarithmic current density.

an increase in EQE and negligible $J_{max\ EQE}$ shift can be further achieved by relatively simple EBL modification, indicating that balanced carrier injection in μLEDs is highly important. Furthermore, with the sidewall passivation, the resulting EQE has enhanced for QB 10.5 Balanced EBL in all device sizes while maintaining the negligible $J_{max\ EQE}$ shift (see Supplementary Figs. 7–10). Figure 4e shows the EL peak wavelength and FWHM value as a function of current density for devices with different QB thicknesses (See supplementary Fig. 5 for EL spectra at different current densities). The EL peak wavelength and FWHM of QB

10.5 Balanced EBL were similar to QB 10.5. As depicted in Fig. 4f, the *S* parameter of QB 10.5 Balanced EBL has a low value than 2 from low current density, which means that tunneling or surface leakage current is effectively reduced with EBL engineering. Since radiative recombination is a process that involves both electrons and holes, the improvement in the balanced injection of carriers can enhance the radiative recombination as depicted in Fig. 3e, which reduces the excessive population of electrons by emitting carriers, resulting in the decrease of leakage current after the MQW region

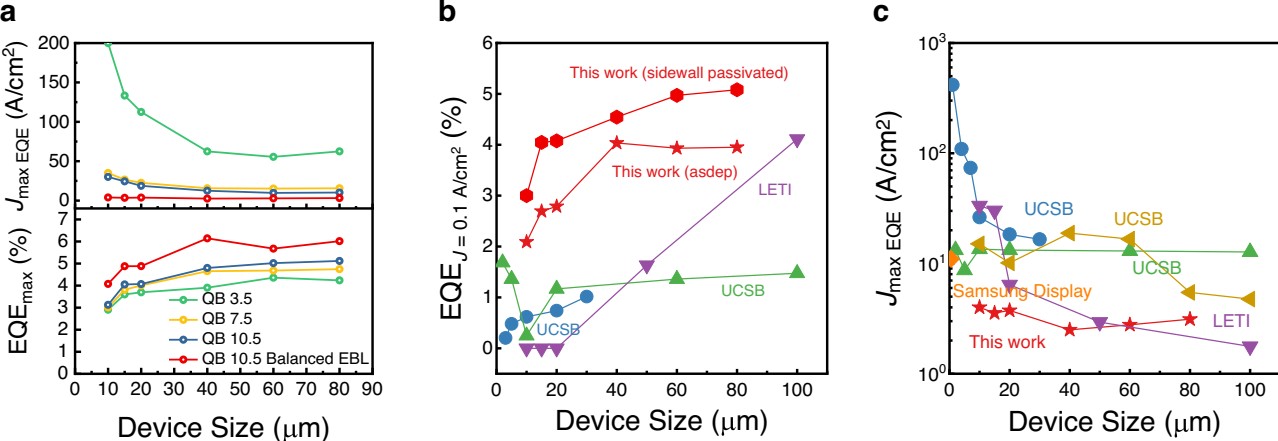

**Fig. 5 | Summary and benchmark of the different epitaxy structures and state-of-the-art blue μLED devices. a** Comparison of maximum external quantum efficiency (EQE) and $J_{max\ EQE}$ of QB 3.5, QB 7.5, QB 10.5, and QB 10.5 Balanced EBL.

Benchmark of **b** EQE at $J = 0.1\ A/cm^2$ and **c** $J_{max\ EQE}$ for state-of-art blue μLED devices as a function of pitch size. (UCSB in blue[4], UCSB in green[5], UCSB in ocher[7], LETI[6], Samsung Display[35]).

(Fig. 3f). For this reason, the lateral diffusion and surface current can be effectively reduced as shown in Fig. 4c.

Figure 5a summarizes the maximum EQE and $J_{max\ EQE}$ of μLED devices fabricated from four epitaxial structures in this paper. Nearly constant $J_{max\ EQE}$ shown in this paper has never been demonstrated in the scaled μLED without surface passivation because previously reported studies have been based on the typical epitaxial design for lighting applications. Figure 5b, c shows the benchmark of EQE at 0.1 A/cm² and $J_{max\ EQE}$ of state-of-the-art μLED devices[4–7,35] (See Supplementary Table 1 in the Supplementary Information for more detail). It shows that μLED devices fabricated from QB 10.5 Balanced EBL show a high EQE of 2.04% (for as-dep) and 3.00% (for sidewall passivated) when the pixel size is $10 \times 10\ \mu m^2$ at the low current density of 0.1 A/cm² and shows that the $J_{max\ EQE}$ is at a low current density with negligible size dependence, which is an important feature for ultra-high resolution μLED display. Here, it should be noted that the decrease of the EQE with size scaling and the low maximum EQE compared to devices grown on the sapphire substrate was attributed to the mask layout hindering light extraction through the metal electrodes and the high absorption coefficient of Si (111) in the visible range of wavelength compared to the sapphire substrate (for $10 \times 10\ \mu m^2$ sized device of QB 10.5 Balanced EBL, a light extraction efficiency of 5.08% was calculated using RTRM method[36]), which leaves a high potential in further enhancements in EQE by increasing the light extraction efficiency.

### Thermal evaluation of μLEDs using thermoreflectance microscopy

The relaxed self-heating effect to prevent temperature increase of pixels during operation is an essential feature for future μLEDs display to guarantee stable display operation to avoid any temperature-dependent unfavorable effects such as wavelength shift. Inefficient LEDs can induce self-heating of the LED device, which can further deteriorate the μLEDs performance. To understand the correlation between the surface effect and the thermal properties of μLEDs fabricated from different epitaxial structures, we measured the temperature distribution of the top surface of μLEDs when the light output power per mesa area (LOP per area) was 1 W/cm² (see Supplementary Table 2 and Supplementary Fig. 12 for more details) with the constant current operation for a device size of $80 \times 80\ \mu m^2$ and $10 \times 10\ \mu m^2$ using CCD-based thermoreflectance microscopy (TRM) system[37]. The measurement method and thermoreflectance coefficient (κ) calibration results can be found in Methods section and Supplementary Figs. 14 and 15. The 2D thermal profile was obtained as shown in

Table 1, and the average temperature of all samples with different sizes is shown in Fig. 6. We found that the average temperature was relatively low in devices with higher EQE and lower current density at corresponding LOP per area. These results indicate that the surface effect, which deteriorates the efficiency of μLEDs, is one of the fundamental reasons for self-heating in μLEDs. Thereby, increasing EQE and mitigating $J_{max\ EQE}$ by alleviating the surface effect through epitaxial engineering can be a fundamental solution to decrease the self-heating of μLEDs in the operating region. Additionally, we found the average temperature of small-sized μLEDs ($10 \times 10\ \mu m^2$) is lower than large-sized μLEDs ($80 \times 80\ \mu m^2$) even though the EQE is lower and the corresponding current density is higher. This may be due to the higher capability of heat dissipation through the pad electrodes suggesting layout design including electrodes will be important to guarantee the thermally stable device operation. The above analysis shows that our epitaxial design of QB 10.5 Balanced EBL successfully decreased the average temperature at typical μLEDs display operating region by increasing the EQE in the low current density region and mitigating the shift of $J_{max\ EQE}$.

In summary, we demonstrated low-current driven μLEDs with high efficiency and self-heating relaxed thermal properties in the low current density by carefully designing QB thickness and EBL. By reducing the tunneling rate of carriers via increasing QB thickness, we successfully increased the confinement of carriers, and reduced the lateral diffusion of carriers and surface current, resulting in the decrease of surface recombination in the sidewall, which eventually increased the EQE and mitigates the shift of $J_{max\ EQE}$ to higher current density region. Furthermore, by modifying the Al composition, doping concentration, and thickness, enhancements in the balance injection of carriers were achieved, which further reduced the surface recombination in the sidewall in the low current density. As a result, we successfully fabricated μLEDs showing a high EQE of 3.00% (sidewall passivated) and at 0.1 A/cm² for the pixel size of $10 \times 10\ \mu m^2$ in GaN-based blue μLEDs and low $J_{max\ EQE}$ throughout the fabricated size with negligible size-dependency. The proposed epitaxial structure showed self-heating relaxed temperature properties during the operation compared to other epitaxial structures, which is a favorable feature in future micro-display.

## Methods
### Epitaxial growth
All four InGaN-based epitaxial structure (QB 3.5, QB 7.5, QB 10.5, and QB 10.5 Balanced EBL) were grown on 8-inch Si (111) wafer in a MOCVD

**Table 1 | Thermal profile of μLEDs with a pitch size of 80 × 80 μm² and 10 × 10 μm² when the light output power (LOP) per area is 1 W/cm² for QB 3.5, QB 7.5, QB 10.5, and QB 10.5 Balanced EBL (BE) samples**

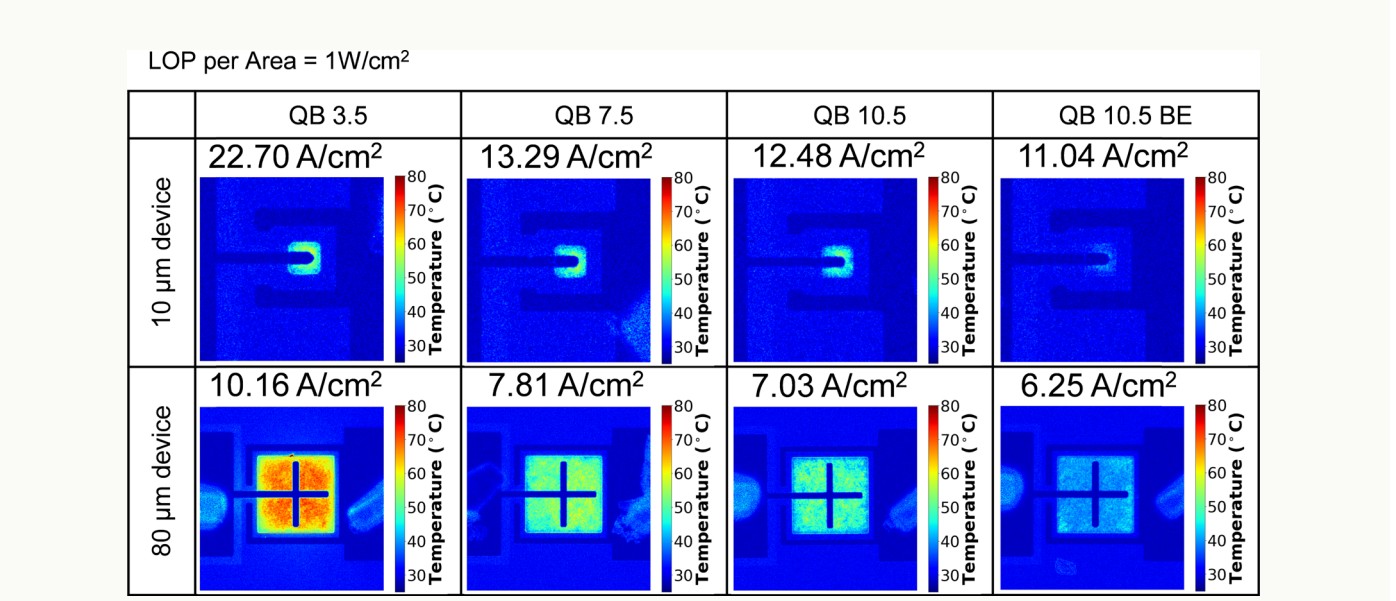

The current density above the thermoreflectance microscopy images shows the corresponding operation condition. The color bar is valid only within the μLEDs mesa region since the thermoreflectance coefficient calibration was conducted for the mesa region.

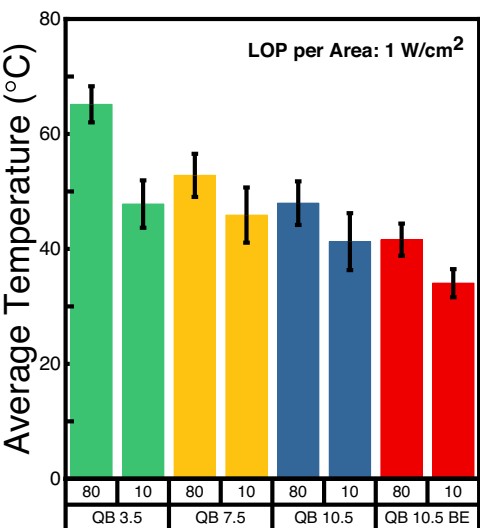

**Fig. 6 | Average temperature of QB 3.5, 7.5, QB 10.5, and QB 10.5 Balanced EBL (BE) with different sizes and light output power (LOP) per area.** The pitch size presented in the figure showing 80 μm and 10 μm corresponds to 80 × 80 μm² and 10 × 10 μm², respectively.

chamber. A 200 nm high-temperature AlN buffer layer is grown. 20 pairs of AlGaN/AlN superlattice layers of thickness ranging from 150 nm to 170 nm are grown, followed by a 150 nm AlGaN buffer layer. Then, an undoped GaN buffer layer with thickness of 1.5 μm, Si-doped n-type GaN with a thickness of 1.5 μm, and 20 pairs of $In_{0.02}GaN/GaN$ (1.5 nm/2 nm) superlattices for strain relief were grown. Then, 6 pairs of InGaN/GaN multiple quantum wells were grown according to their QB thickness specification. For the EBL structure, in the case of QB thickness-controlled samples (QB 3.5, QB 7.5, and QB 10.5), a 17 nm thick $Al_{0.2}In_{0.02}Ga_{0.78}N$ with $2×10^{18}$ cm⁻³ Mg p-type doping concentration was grown, and in case of QB 10.5 Balanced EBL, a 45 nm thick $Al_{0.12}In_{0.02}Ga_{0.86}N$ with $2×10^{19}$ cm⁻³ Mg p-type doping

concentration was grown. At last, highly Mg-doped p-type layer and a very thin InGaN capping layer were grown.

**Device fabrication**

After cleaning with acetone/methanol/deionized water in a sonication bath, the samples were cleaned using an $H_2SO_4{:}H_2O_2$ solution to remove the organic residue. Then, HCl: deionized water = 1:1 (v/v) solution was used to remove the native oxide and was loaded immediately into an e-beam evaporator in order to deposit indium tin oxide (ITO) transparent contact metal. After the deposition, the samples were annealed using a rapid thermal annealing process for 5 min at 550 °C in air ambient. $SiO_2$ layer was deposited by plasma-enhanced chemical vapor deposition (PECVD) process above ITO layer to form a etch stop layer for mesa formation. A positive photoresist (PR) was patterned on the $SiO_2$ layer and buffered oxide etch (BOE) solution etched the $SiO_2$ layer to define the precise mesa area. Consequently, $Cl_2/N_2$ gas mixture based inductively coupled plasma-reactive ion etch (ICP-RIE) was used to etch down to the n-GaN layer. For as-deposited devices, sidewall passivation of mesa such as KOH passivation is not conducted in order to observe the effect of epitaxial structure on device performance, and the default devices are as-deposited device unless mentioned. After the etching, the $SiO_2$ mask was etched with buffer oxide etch solution. For the devices with the sidewall passivation, the μLEDs were treated with KOH (2 mol/L) at room temperature for 40 min after the ICP-RIE etching. After the sidewall passivation, the $SiO_2$ hard mask is removed and sequentially loaded into the ALD chamber for ALD passivation. 50 nm-thick $Al_2O_3$ layer was deposited using atomic layer deposition and was patterned and etched to form insulating areas for p-contact metal. Ti/Au (20/200 nm) was deposited using an e-beam evaporator and was lifted off using acetone. Regarding the history of chemicals and fabrication processes exposed to the sidewall, the device sidewall is exposed to buffered oxide etch solution for 2 min when removing the $SiO_2$ hard mask and is covered by $Al_2O_3$ through atomic layer deposition during the formation of isolation layer for p-type contact. The covered $Al_2O_3$ is then completely etched. During the metal pad lithography process, the sidewall is exposed to DNR-L300-40 which is a negative photoresist, and at last,

acetone is exposed to the sidewall during the liftoff process. The device fabrication schematic is in Supplementary Fig. 13.

## Measurement method

The electrical characteristic of µLEDs was measured using Keysight B1500A. The optical characteristics were measured with a 4-inch integrated sphere (IS) and Si-photodiode (PD) (Hamamatsu S1337-21). The photoluminescence spectrum was obtained using LabRAM HR Evolution Visible-NIR, with a Hd-Cd laser wavelength of 325 nm at room temperature. The distance from the samples to the laser was kept identical when measuring all samples. TEM images were obtained by JEM-2100F, and the EDS data were obtained by X-Max$^N$ from Oxford Instruments. To quantitatively acquire the temperature profile of µLEDs operating at different light output power and current density, TRM was used to obtain the thermal image by biasing devices with a modulated voltage of 2 Hz, and the image was recorded with the frame rate of 30 Hz for 15 s. In order to separate the reflectance variation caused by the light emission and the thermal effect, the CCD camera (Pco Edge 4.2 LT) was aligned with a bandpass filter with the wavelength of 586 nm (with FWHM of ~10 nm), which can eliminate the reflectance change due to the blue light emission. The thermoreflectance coefficient was obtained by measuring and calibrating the change of averaged reflectance value of mesa area with a step-increased temperature of thermoelectric cooler element. The setup schematic is shown in Supplementary Fig. 14. A modification in the TRM setup was conducted for EL mapping. The bandpass filter with the wavelength of 586 nm was removed so that the CCD can detect the blue light wavelength. The EL mapping was obtained by modulating the device with a 2 Hz voltage pulse train which corresponds to 0.1 A/cm$^2$.

## Simulation

The band diagram, carrier population, and recombination rate are numerically investigated using Silvaco Atlas. The simulated LED structure used is identical to the samples proposed in this study but was simulated in a vertical structure. The device geometry was fixed to $10 \times 10$ µm$^2$ for all simulations. The interface charge densities induced by spontaneous and piezoelectric polarization were assumed to be 50% compared to the theoretical polarization. The band offset ratio is assumed to be $\Delta E_c : \Delta E_v = 0.5 : 0.5$[33]. The surface recombination velocity was set to $1.0 \times 10^5$ cm/s. The SRH recombination lifetime, radiative recombination and Auger recombination coefficient were set to 100 ns, $1 \times 10^{-10}$ cm$^3$/s, and $1 \times 10^{-31}$ cm$^6$/s, respectively. Other material parameters are similar to those employed in literature[38].

## Data availability

The datasets generated during and analyzed during this study are available from the corresponding author upon request.

## Code availability

The code used for the simulation in this study are not publicly available because authors do not have the right to publish the base codes provided by Silvaco Inc. which are included in the simulation code. However, they are available from the corresponding author upon reasonable request.

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

## Acknowledgements
This work was supported by Samsung Research Funding & Incubation Center of Samsung Electronics under project number SRFC-TB2003-02 (W. B, J. P, J. S, H. K, S. K), BrainKorea (BK21) FOUR, and IDEC.

## Author contributions
W.B. conducted the device fabrications, characterizations, and analysis. J.P., J.S., B.K., S.P., H.K., and D.G. contributed in the experiment and characterization methodology. S.K. and D.G. jointly supervised the project. W.B. wrote the manuscript.

## Competing interests
The authors declare no competing interests.
