## [Peer Review File · Nature Communications]

Ultra-low-current Driven InGaN Blue Micro Light-emitting Diodes for Electrically Efficient and Self-heating Relaxed MicrodisplayEditorial Note: Parts of this Peer Review File have been redacted as indicated to maintain the confidentiality of unpublished data.

REVIEWER COMMENTS

Reviewer #1 (Remarks to the Author):

The manuscript entitled “Ultra-Low-Current driven InGaN blue micro light-emitting diodes for electrically efficient and self-heating relaxed microdisplay” presents new LED structure designs to shift the maximum external quantum efficiency (EQE) to low current density and to reduce the self heating of the micro-LEDs under operation. Two bricks in the structure are considered: the thickness of the quantum barrier (QB) and the design of the electron blocking layer. Micro-LEDs from sizes of $10 \times 10 \mu\text{m}^2$ to $80 \times 80 \mu\text{m}^2$ were fabricated and measured for all the structure variations. The aim of the study was to demonstrate that the EQE degradation versus micro-LED size can be reduced by playing on the LED structure and thus by adapting the LED structure to the specifications of $10 \times 10 \mu\text{m}^2$ micro-LEDs for micro-display applications. And thus not by adding only passivation of the micro-LED sidewalls. While the results are very interesting, I recommend major revisions as some conclusions should be more clarified. You will find my comments below.

- The figures are too thin, especially the TEM images, and not all the figures are fully commented. Is it possible to make a choice in the presented figures and/or to make a specific figure for the TEM images of Fig. 1? All the figures should be fully commented.
- The use of a QB at 3.5 nm is surprising. With such a thin QB, there should be an overlap of the e-h wavefunctions in the QB, so a reduction of the radiative recombinations in the quantum wells (QWs). And as there is a redistribution of the internal electric field between the QW and the QB, there is a strong blueshift of the emission. This is what is observed in Fig. 1(f). It is not dependant of the size of the device and this would have been more anticipated as this is known. In addition, due to the strong blue shift the micro-LEDs with QB at 3.5 nm can not be compared with the others in term of EQE. Please modify the text and conclusions accordingly.
- It is claimed that there is a moderate dependency of the EQE versus the device size for the two best structures. However, this quite moderate dependency is observed for all the LED structures and is not dependent here on the LED structure variations. This should be commented and modified in the whole text.
- The result of Fig. 2(b) is very interesting and should be more highlighted in the figures, while it is well explained in the text pages 8-9. Would it be possible to perform electroluminescence mapping of the different micro-LEDs to visualize more this result? This is actually what I was expected by reading the abstract. It is indeed written “efficiency degradations by reducing the lateral diffusion of the injected carriers”. This should give stronger impact on these results.
- Fig. 3(a): for the LED structure of QB 10.5, the spacer is too thick. The effect of the new EBL would have been the same between the two structures if the spacer has been thinner? Can you comment on this in the text?
- Finally, most of the radiative recombinations are located in the last QW near the p type region as this is usually the case for standard blue LEDs (Fig. 3(e)). Can you comment on this? Is it really new for micro-LEDs? Do you take into account in your simulations the surface recombinations at the micro-LED sidewalls?

- Fig. 4(b): why showing JS/JB(V) ? While it is only JS(V) for Fig. 2(b).
- Fig. 4(e) not necessary.
- General comment on EQEmax values: The final current density at EQEmax obtained with the best structure (3 A/cm²) has a value equivalent as standard blue LED (using structure without pushing too much the droop to higher current density). Can you comment on this? Can you also comment on the fact that the EQEmax for all the structures does not change too much (between 2.5 and 4%)?
- Device fabrication method: can you precise in the text that there is no passivation of the micro-LED sidewalls? Can you also precise the growth technique?
- Measurement method: there is no mention of the EDX measurements.

Reviewer #2 (Remarks to the Author):

This manuscript reports the fabrication of μ LEDs with high emission efficiency and good thermal properties using an optimized design of QB thickness and EBL. The resulting μ LED devices exhibit a high EQE of 2.04% at 0.1 A/cm² for $10 \times 10 \mu\text{m}^2$ sizes. This result suggests that a new epitaxial approach can effectively reduce the μ LED pixel temperature, which can contribute to the production of high-resolution μ LED displays. The manuscript, however, includes unclear results and descriptions. Before the final decision of publication, the authors should address the following comments.

1. Wong et al, Appl. Phys. Express 12, 097004 (2019) and Sheen et al, Nature 608, 56 (2022) report much higher EQE for μ LEDs of $< 10 \times 10 \mu\text{m}^2$ size than that of this work. The authors should compare with the results of previous publications at fixed current density. "record high" word cannot be used without a clear analysis and comparison with the previous reports.
2. If the side wall passivation is properly used, it would benefit EQE. It is recommended that the authors add the data related to the change in the thermal and emission properties induced by the side wall passivation.
3. The authors demonstrate $10\mu\text{m} \sim 80\mu\text{m}$ μ LEDs. Recently, μ LEDs with a size of less than 5 μm have been produced for high-resolution display. Therefore, it would be important to show thermal properties and EQE characteristic data for μ LED with less than 5 μm .

Reviewer #3 (Remarks to the Author):

This article optimizes the performance of micro-LEDs at low current densities through epitaxial structure design and analyzes the underlying physical mechanisms. However, there has been a lot of research on the adjustment and analysis of the epitaxial structure of LED devices, and this article is not novel enough in this regard. What's more, the article has the following problems, which need to be further revised.

- 1: What is the selection principle of quantum barrier thickness in this work? As shown in the study, the performance of small-scale micro-LED is progressively improved as the thickness of the quantum barrier

increases from 3.5 nm to 10.5 nm. But for thinner or thicker quantum barrier thicknesses, the author did not mention. Please clarify.

2: By optimizing the quantum barrier thickness and EBL structure, the authors obtained a relatively higher EQE at ultra-low current densities, which is very attractive for micro-display. However, the peak EQE value of the blue micro-LED designed by this study is much lower than the reported researches [1], and even the $80 \times 80 \mu\text{m}^2$ device, the peak EQE is only $\sim 6\%$. What are the differences among them? In other words, are the optimization methods of this work applicable to different epitaxy designs?

[1] Wong, Matthew S., et al. "Size-independent peak efficiency of III-nitride micro-light-emitting-diodes using chemical treatment and sidewall passivation." *Applied Physics Express* 12.9 (2019): 097004.

3: In Fig. 1(f), it can be seen that the quantum barrier thickness has a huge effect on the PL wavelength. The authors should present EL spectra of different micro-LEDs at different current densities, which are important for actual display devices.

4: How do the authors consider the potential effect of quantum well thickness on EQE and J_{max} EQE? Is there an optimized principle for thickness selection of both QW and QB?

5: Some details need to be carefully checked and revised, such as Fig. 2(c) and (f) are exactly the same.

6: The authors should provide the average temperature data at different light output power densities (the corresponding current densities should be clearly given) to better study the difference between different epitaxial structures.

7: In this paper, the junction current and sidewall current with different QB are analyzed in detail. However, for actual device fabrication, sidewall passivation has a great influence on the suppression of non-radiative recombination and should not be neglected. It is not known whether the sidewall passivation will affect the results of these analyses. Please clarify.

8. For the same QB thickness, J_{max} EQE increases with the decrease in size. And for the same size, J_{max} EQE decreases with the increase of QB thickness. More reasons need to be given.

Response Letter to Reviewer's Comments

on “Ultra-Low-Current Driven InGaN Blue Micro Light-Emitting Diodes for Electrically Efficient and Self-Heating Relaxed Microdisplay”

Reviewer's Comment: blue; Responses: black; Changes made: highlighted;

The line numbers are for the manuscript with highlighted changes.

[General Response]

We would like to express our sincere appreciation for the valuable time spent by the reviewers and the editor for reviewing our manuscript and providing insightful comments and constructive suggestions to help further enhance the quality and clarity of our work. According to the reviewers' comments, we have integrated new data from additional work and modified the manuscript and Supplementary Information.

Reviewer #1 (Remarks to the Author):

[Reviewer's Comment]

The manuscript entitled “Ultra-Low-Current driven InGaN blue micro light-emitting diodes for electrically efficient and self-heating relaxed microdisplay” presents new LED structure designs to shift the maximum external quantum efficiency (EQE) to low current density and to reduce the self-heating of the micro-LEDs under operation. Two bricks in the structure are considered: the thickness of the quantum barrier (QB) and the design of the electron blocking layer. Micro-LEDs from sizes of $10 \times 10 \mu\text{m}^2$ to $80 \times 80 \mu\text{m}^2$ were fabricated and measured for all the structure variations. The aim of the study was to demonstrate that the EQE degradation versus micro-LED size can be reduced by playing on the LED structure and thus by adapting the LED structure to the specifications of $10 \times 10 \mu\text{m}^2$ micro-LEDs for micro-display applications. And thus not by adding only passivation of the micro-LED sidewalls.

While the results are very interesting, I recommend major revisions as some conclusions should be more clarified. You will find my comments below.

[Response]

We sincerely appreciate reviewer #1 for positive comments on our revised manuscript. We thank you for valuing our work of epitaxial structure engineering achieving the mitigation of size-dependent degradation of μLEDs even without the sidewall passivation. We additionally elaborated on the performance of devices with sidewall passivation. The performance has enhanced in all devices with four different epitaxial structures, but the trend remains the same, emphasizing that epitaxial engineering is not an option but a must for μLEDs display. The responses to your comment can be found below.

[Reviewer's Comment]

1. The figures are too thin, especially the TEM images, and not all the figures are fully commented. Is it possible to make a choice in the presented figures and/or to make a specific figure for the TEM images of Fig. 1? All the figures should be fully commented.

[Response]

We appreciate the constructive comment on the visibility and the lack of a full explanation on some figures. Based on the reviewer's comments and suggestions, we have increased the size of the TEM images (Fig. R1) in the manuscript and added bigger TEM images (Fig. R2) of the epitaxial structure in the supplementary figure. Furthermore, figures with a lack of explanations are added as well.

Fig. R1 Transmission electron microscope (TEM) images of grown epitaxy layers. **a** from superlattice layers to p-GaN of QB 3.5. **b-d** Active region of **b** QB 3.5 **c** QB 7.5 **d** QB 10.5 The red line in **b-d** shows the line-scanned energy dispersive spectroscopy of the indium component.

Fig. R2 Transmission microscope image of QB 3.5, QB 7.5, QB 10.5, and QB 10.5 Balanced EBL from the superlattice layers to the p-GaN layer. a QB 3.5 b QB 7.5 c QB 10.5 d QB 10.5 Balanced EBL.

Modified Content

Fig. R1 is updated to Fig. 1b-e in the main manuscript.

Fig. R2 is added to Supplementary Information.

[Legend of Fig. 1] **Fig. 1 Characterization of grown μ LED epitaxy for the redesign of quantum well barrier in μ LED.** a schematic of epitaxy structure and designated active region structure with different QB thicknesses. b-e transmission electron microscope (TEM) images of grown epitaxy layers (b from superlattice layers to p-GaN of QB 3.5. c-e Active region of c QB 3.5 d QB 7.5 e QB 10.5) The red line in c-e shows the line-scanned energy disperse spectroscopy of indium component. f photoluminescence (PL) spectra of QB 3.5, QB 7.5, and QB 10.5. g schematic of fabricated devices. h top-view optical microscope image of fabricated $10 \times 10 \mu\text{m}^2$ μ LED devices without and with injection current. i scanning electron

microscope (SEM) images of the fabricated μ LEDs in different sizes.

[Legend of Fig. 2] **Fig. 2 Electrical and optical characteristics of QB 3.5, QB 7.5, and QB 10.5.** **a** Logarithmic J - V characteristic of μ LEDs with a pitch size of $80 \times 80 \mu\text{m}^2$ and $10 \times 10 \mu\text{m}^2$. **b** J_S and **c** J_S/J_B as a function of voltage for three different samples. (inset of **b**. Schematic image of PIN LEDs and the components of forward current paths.) **d** EQE-logarithmic current density curve for QB 3.5, QB 7.5, and QB 10.5 with pitch sizes from $80 \times 80 \mu\text{m}^2$ to $10 \times 10 \mu\text{m}^2$. (The green line shows the point where the $10 \times 10 \mu\text{m}^2$ device is at its maximum EQE) **e** $J_{max\ EQE}$ and maximum EQE of devices with different sizes. **f** electroluminescence peak wavelength and FWHM. **g** S parameter as a function of logarithmic current density. **h** Schematic diagram of possible carrier transport processes in μ LEDs under forward bias. The electrons drifting or tunneling throughout the active region can diffuse to the sidewall where the surface recombination can occur or drift through the surface leakage path. The drifted or tunneled electrons through the active region can also accumulate at the interface between the active region and the electron blocking layer, where a lateral diffusion can be significant. **i** normalized electroluminescence intensity mapping of QB 3.5, QB 7.5, and QB 10.5. The pitch size of the device is $80 \times 80 \mu\text{m}^2$, and the operating current density is 0.1 A/cm^2 .

[Legend of Fig. 4] **Fig. 4 Electrical and optical characteristics of QB 10.5 and QB 10.5 Balanced EBL.** **a** logarithmic J - V characteristic of QB 10.5 and QB 10.5 Balanced EBL with pitch size of $80 \times 80 \mu\text{m}^2$ and $10 \times 10 \mu\text{m}^2$. (inset. normalized electroluminescence wavelength of μ LEDs with a pitch size of $80 \times 80 \mu\text{m}^2$ driven at 1mA .) **b** J_S as a function of voltage **c** J_S/J_B as a function of voltage. **d** EQE-logarithmic current density curve for QB 10.5 and QB 10.5 Balanced EBL with pitch size from $80 \times 80 \mu\text{m}^2$ to $10 \times 10 \mu\text{m}^2$. (The green line shows the point where the $10 \times 10 \mu\text{m}^2$ device is at its maximum EQE) **e** electroluminescence peak wavelength and FWHM. **f** S parameter as a function of logarithmic current density. **e** μ LEDs images of QB 10.5 Balanced EBL driven under different current density

[Legend of Table 1] **Table 1. Thermal profile of μ LEDs with a pitch size of $80 \times 80 \mu\text{m}^2$ and $10 \times 10 \mu\text{m}^2$ when the light output power (LOP) per area is 1 W/cm^2 for QB 3.5, QB 7.5, QB 10.5, and QB 10.5 Balanced EBL (BE) samples.** The current density above the thermoreflectance microscopy images shows the corresponding operation condition. The scale bar is valid only within the μ LEDs mesa region since the thermoreflectance coefficient calibration was conducted for the mesa region

[Reviewer's Comment]

2. The use of a QB at 3.5 nm is surprising. With such a thin QB, there should be an overlap of the e-h wavefunctions in the QB, so a reduction of the radiative recombinations in the quantum wells (QWs). And as there is a redistribution of the internal electric field between the QW and the QB, there is a strong blueshift of the emission. This is what is observed in Fig. 1(f). It is not dependant of the size of the device and this would have been more anticipated as this is known. In addition, due to the strong blue shift the micro-LEDs with QB at 3.5 nm can not be compared with the others in term of EQE. Please modify the text and conclusions accordingly.

[Response]

We appreciate reviewer #1 for this constructive comment. We agree with the comment that there should be an overlap of e-h wavefunctions in the QB if the QB thickness is thin, which can result in the decrease of radiative recombination. This must be considered when designing a QB thickness of multiple quantum wells. On the other hand, from a different point of view, there are several papers stating that the decreased QB thickness will benefit radiative recombination due to the decreased internal field in the QW, which will eventually enhance the overlap between the electron and hole wavefunction¹⁻³. However, what they didn't concerned about is the redistribution of internal electric fields which can influence the wavefunctions in the neighboring QWs, which you have commented on. In fact, even though they have stated that the thin QB can enhance the efficiency droop, the resulting EQE of the thin QBs in their experimental data was lower than that of the thicker QBs.

Relating your comment to our manuscript, we think that the decrease of radiative recombination due to the overlap of e-h wavefunctions in the QB is also the reason for the increase of lateral diffusion of carriers when QB thickness is thinner. In MQWs with a thin barrier thickness which we normally call superlattice, a coupling of wavefunctions between adjacent QWs forms minibands of carriers. The carriers can transport across the superlattice layer without being recombined⁴. In the epitaxial structure of barrier thickness of 3.5 nm, we can expect miniband formation since the strong blue shift in the photoluminescence spectra. For this reason, the radiative recombination in the formed miniband can accelerate the drift of carriers without being recombined, which eventually increases the lateral diffusion, resulting in the decrease of EQE. Nevertheless, our claim is not to use a thinner QB structure, but to show the epitaxy strategy to increase the QB thickness. Therefore, the novelty of this paper is still strong and provides insight into scaled μ LEDs, which has not been discussed before. To address the reviewer's concern, we have changed the manuscript as follows.

Modified Content

[Line 98] In fact, besides the wavefunction and the kinetic energy, QB thickness scaling would also lead to the formation of the miniband, change in barrier height in the QW energy conduction band and the tunneling rate, which must be taken into account in the epitaxial structure design in μ LEDs.

[Line 134] In addition, it is worth noting that the decrease of PL intensity and strong blue shift of peak wavelength were observed in with the scaled QB thickness of 3.5 nm, indicating relatively low radiative recombination compared to QB 7.5 and QB 10.5 samples. This is because of the formation of miniband due to the coupling of wavefunctions between adjacent

QWs. The formation of miniband can result in the strong blueshift of PL spectra and the reduction in radiative recombination⁴.

[Line 274] Eventually, the lateral diffusion of carriers and surface current can cause a significant reduction in EQE due to the increased non-radiative surface recombination throughout the mesa sidewall. Furthermore, the formation of miniband in QB 3.5 can accelerate the lateral diffusion of carriers, since the carriers can tunnel through the miniband without being recombined⁴. As a result, the amount of $J_{max\ EQE}$ shift to the high current density region was much higher when the device size decreased and the maximum EQE was smaller in μ LEDs with thinner QBs as depicted in Fig. 2e

[Reference] Nakamura, T. et al. Effect of Built-in Electric Field on Miniband Structure and Carrier Nonradiative Recombination in InGaAs/GaAsP Superlattice Investigated Using Photoreflectance and Photoluminescence Spectroscopy. *Energy Procedia* **102**, 121-125 (2016).

Regarding the EQE, we have calculated the EQE according to equation (1):

$$\eta_{EQE} = \frac{\int_0^{\infty} \frac{\lambda}{hc} \frac{d\Phi_e(\lambda)}{d\lambda} d\lambda}{I/q} \quad (1)$$

where λ is the wavelength of the emitted light, h is the Planck constant, c is the speed of light in vacuum, Φ_e is the radiant power emitted, I is the forward current injected, and q is the elementary charge.

The shift of the wavelength of emitted light has been accounted for the EQE, which has been calculated to the number of photons emitted per time. Since we are comparing photons emitted per time over the injected carrier per time, we think that the EQE comparison is valid.

[Reviewer's Comment]

3. It is claimed that there is a moderate dependency of the EQE versus the device size for the two best structures. However, this quite moderate dependency is observed for all the LED structures and is not dependent here on the LED structure variations. This should be commented and modified in the whole text.

[Response]

Thank you for this comment. We admit that there are ambiguous statements in terms of size dependency. Our intention of the statements regarding the moderate size dependency is for the shift of current density showing maximum EQE ($J_{max\ EQE}$) in QB 10.5 Balanced EBL, and we want to emphasize that there is no direct statement stating that the EQE has moderate size dependency.

To avoid confusion, we have changed the ambiguous statements.

Modified Content

[Abstract – Line 25] As a result, the fabricated micro-light-emitting diodes μ LEDs show a record-high external quantum efficiency EQE of 3.00% at 0.1 A/cm² for the pixel size of 10 × 10 μ m² and a negligible size dependency $J_{max\ EQE}$ shift during size reduction, which is

extremely challenging due to the non-radiative recombination at the sidewall.

[Reviewer's Comment]

4. The result of Fig. 2(b) is very interesting and should be more highlighted in the figures, while it is well explained in the text pages 8-9. Would it be possible to perform electroluminescence mapping of the different micro-LEDs to visualize more this result? This is actually what I was expected by reading the abstract. It is indeed written “efficiency degradations by reducing the lateral diffusion of the injected carriers”. This should give stronger impact on these results.

[Response]

We sincerely thank reviewer #1 for valuing our results in finding the difference in surface current induced by different epitaxial structures. Our attempt to highlight this result was by plotting J_S/J_B - V curve shown in Fig. 2c in the original manuscript, which we have modified in the revised version as you have pointed out in your comment #7. In our opinion, showing the J_S/J_B - V curve can highlight the results of surface current in terms of quantitative measures.

Furthermore, we also found that the trend in Fig. 2b (which is the same as Fig. R3a) showing the increasing surface current with decreased QB thickness was still valid when the sidewall passivation (KOH treatment + Al₂O₃ passivation) is conducted. Fig. R3 shows the J_S and J_S/J_B curve as a function of voltage for as-deposited (asdep) and sidewall passivated devices. While the increasing trend of J_S and J_S/J_B for decreasing QB thickness was identical after the sidewall passivation. A decrease in the surface current was observed even after the sidewall passivation, strongly convincing the validity of the surface current analysis method and the hypothesis of this paper, which the lateral diffusion of carriers can be reduced by epitaxial engineering.

Fig. R3 J_S and J_S/J_B curve as a function of voltage for the as-deposited (asdep) and sidewall passivated μ LEDs devices. **a** J_S -V curve **b** J_S/J_B -V curve for asdep devices. **c** J_S -V curve **d** J_S/J_B -V curve for sidewall passivated devices.

Regarding the visual highlight of the findings, we have mapped the electroluminescence (EL) intensity by modifying the thermoreflectance microscopy as shown in Fig. R4. From the figure, it is possible to observe a lower normalized EL intensity at the surfaces near the sidewall in QB 3.5. A similar finding was also observed in QB 7.5, but the low normalized EL intensity near the sidewall was less severe than in QB 3.5. In the case of QB 10.5, a uniform EL intensity was observed throughout the device region. This was indeed quite interesting and we think that this should be highlighted to support our findings. We would like to appreciate the reviewer's comment. For higher current density, we could not obtain the image due to the blooming effect in the CCD. For the devices with a size of $10 \times 10 \mu\text{m}^2$, this effect was hard to observe due to the high background noise.

Fig. R4 Normalized electroluminescence intensity mapping of QB 3.5, 7.5, and QB 10.5. The pitch size of the device is $80 \times 80 \mu\text{m}^2$, and the operating current density is 0.1 A/cm^2 .

Modified Content

Fig. R3c-d and Fig. R4 are added to Supplementary Information and Fig. 2i, respectively. The corresponding explanation and measurement details were added in the main manuscript, which is as below:

[In the Supplementary Information] In order to observe the impact of sidewall passivation on the surface current, we fabricated the device with the sidewall passivation scheme. Supplementary Fig. 7 shows the J - V characteristics of differently sized μLEDs with sidewall passivation. Comparing the J - V characteristics with Supplementary Fig. 3 which shows the results from asdep devices, the gradual increase of current density was reduced dramatically as shown in Supplementary Fig. 7e-f. The J_S - V curve and J_S/J_B - V curve are shown in Supplementary Fig. 8 to compare the surface current. We observe a decrease in both J_S and J_S/J_B as shown in Supplementary Fig. 8a-b. It is worth mentioning that the trend of decrease of J_S/J_B when the QB thickness increases is still valid even though the sidewall passivation has been conducted, thus the trend in QB 10.5 and QB 10.5 Balanced EBL. Supplementary Fig. 8c-f shows the S parameter of the asdep and sidewall passivated device with a size of $10 \times 10 \mu\text{m}^2$. From the figure, a decrease in the S parameter was observed especially in the low current density for all structures. However, while the S parameters for thin QB thickness have more a dramatic decrease such as QB 3.5, a smaller change occurred in QB 10.5 Balanced EBL, emphasizing that QB 10.5 Balanced EBL has more immunity to the sidewall efficiency degradation due to its epitaxy nature of less sidewall current.

[Line 284] Additionally, to visualize the effect of surface current in devices with different QB thicknesses, the electroluminescence (EL) intensity mapping was conducted as shown in Fig. 2i. From the figure, it is possible to observe a lower normalized EL intensity at the surfaces near the sidewall in QB 3.5. A similar finding was also observed in QB 7.5, but the low normalized EL intensity near the sidewall was less severe than in QB 3.5. In the case of QB 10.5, a uniform EL intensity was observed throughout the device region.

[Methods – Measurement method – Line 531] A modification in the TRM setup was conducted

for EL mapping. The bandpass filter with the wavelength of 586 nm was removed so that the CCD can detect the blue light wavelength. The EL mapping was obtained by modulating the device with a 2 Hz voltage pulse train which corresponds to 0.1 A/cm^2 .

[Reviewer's Comment]

5. Fig. 3(a): for the LED structure of QB 10.5, the spacer is too thick. The effect of the new EBL would have been the same between the two structures if the spacer has been thinner? Can you comment on this in the text?

[Response]

Thank you for this important comment. In order to respond to your comment, we have conducted additional TCAD simulations of two different structures:

1. QB 10.5 w/o spacer, where the structure is identical to QB 10.5 but without the spacer.
2. QB 10.5 Balanced EBL w/ spacer, where the structure is identical to QB 10.5 Balanced EBL, but there is a spacer layer between the last QB and EBL.

The effect of the spacer depends on the EBL. The spacer acts as a layer that compensates the electric field caused by the piezoelectric difference between the last quantum barrier and the electron blocking layer which has a high aluminum composition ($\text{Al}_{0.2}\text{In}_{0.02}\text{Ga}_{0.78}\text{N}$). Comparing the energy band diagram of QB 10.5 and QB 10.5 w/o spacer, due to the piezoelectric difference and limited doping concentration caused by high Al composition, a deeper energy valley is formed in the interface between the last QB and EBL in the case of QB 10.5 w/o spacer as shown in Fig. R5a. As a result, the leakage current of QB 10.5 w/o spacer is higher as depicted in Fig. R5b. This increased accumulation of electrons and leakage current can increase the lateral diffusion of electrons to the sidewall in the low current region, which we have shown through the correlation between the simulation and experimental results of QB 10.5 and QB 10.5 Balanced EBL in the original manuscript.

However, through the simulation, we found that the importance of the spacer in QB 10.5 Balanced EBL is not as significant as in QB 10.5. Fig. R6 shows the simulation results of QB 10.5 Balanced EBL (same as the results in Fig. 3 of the original manuscript) and QB 10.5 Balanced EBL w/ spacer. Due to the lowered Al composition and induced higher p-type doping concentration of EBL in QB 10.5 Balanced EBL, the energy valley is not formed in the interface between the last QB and EBL as shown in Fig. R6a, which means the necessity to compensate the electric field between this interface is not significant. The simulated leakage current results in Fig. R6b also show that there is no significant difference between the leakage current. For this reason, if there is no spacer in QB 10.5, the effect will not be the same as the QB 10.5 Balanced EBL, rather the performance will be expected to be worse than QB 10.5. In the meanwhile, the spacer is not as significant in QB 10.5 Balanced EBL. In other words, rather than the effect of the spacer itself, the effect of the spacer is highly dependent on the structure of EBL.

Fig. R5 Simulation results of QB 10.5 and QB 10.5 w/o spacer. a energy band diagram **b** electron current density. The simulations are at the current density of 0.1 A/cm^2 .

Fig. R6 Simulation results of QB 10.5 Balanced EBL and QB 10.5 Balanced EBL w/ spacer. a energy band diagram **b** electron current density. The simulations are at the current density of 0.1 A/cm^2 .

Modified Content

Fig. R5, Fig. R6, and the corresponding explanation are added to the Supplementary Information.

[In the Supplementary Information] In order to show the role of the spacer and the reason why no spacer exists in QB 10.5 Balanced EBL, simulations of two additional structure is conducted:

1. QB 10.5 w/o spacer, where the structure is identical to QB 10.5 but without the spacer.
2. QB 10.5 Balanced EBL w/ spacer, where the structure is identical to QB 10.5 Balanced EBL,

but there is a spacer layer between the last QB and EBL.

The existence of a spacer is important in QB 3.5, QB 7.5, and QB 10.5. It acts as a layer that compensates the electric field caused by the piezoelectric difference between the last quantum barrier and the electron blocking layer which has a high aluminum composition ($\text{Al}_{0.2}\text{In}_{0.02}\text{Ga}_{0.78}\text{N}$). Comparing the energy band diagram of QB 10.5 and QB 10.5 w/o spacer, due to the piezoelectric difference and limited doping concentration caused by high Al composition, a deeper energy dip is formed in the interface between the last QB and EBL in the case of QB 10.5 w/o spacer as shown in Supplementary Fig. 11a. As a result, the leakage current of QB 10.5 w/o spacer is higher as depicted in Supplementary Fig. 11b. This increased accumulation of electrons and leakage current can increase the lateral diffusion of electrons to the sidewall in the low current region.

However, through the simulation, we found that the importance of the spacer in QB 10.5 Balanced EBL is not as significant as in QB 10.5. Supplementary Fig. 11c shows the energy band diagram QB 10.5 Balanced EBL and QB 10.5 Balanced EBL w/ spacer. Due to the lowered Al composition and induced higher p-type doping concentration of EBL in QB 10.5 Balanced EBL, the energy dip is not formed in the interface between the last QB and EBL in the energy band diagram, which means the necessity to compensate the electric field between this interface is not significant. The simulated leakage current results in Supplementary Fig. 11d also show that there is no significant difference between the leakage current.

[Reviewer's Comment]

6. Finally, most of the radiative recombinations are located in the last QW near the p type region as this is usually the case for standard blue LEDs (Fig. 3(e)). Can you comment on this? Is it really new for micro-LEDs? Do you take into account in your simulations the surface recombinations at the micro-LED sidewalls?

[Response]

We express our appreciation for the valuable comment. As you have commented, it is commonly known that the majority of radiative recombination occurs in the last QW near the p-type region for standard blue LEDs. However, the significance of simulation figures is not the dominance of radiative recombination in the last QW. What we wanted to emphasize through Fig. 3 (you can find the same figure in Fig. R6) in the manuscript is the importance of the balanced injection of carriers and the resulting leakage current, which finally impacts the EQE at the low current range in scaled LEDs. More importantly, we wanted to show how this leakage current can be attributed to the surface current in experimental data shown in Fig. 4. Moreover, the importance of this study in terms of the electron blocking layer is that this newly designed EBL (QB 10.5 Balanced EBL) will not be used in LEDs for general lighting purposes due to the severe efficiency droop in the high current density region, but this epitaxy of QB 10.5 Balanced EBL is the most suitable one among all four in terms of display purpose due to its high efficiency in the low current density region and negligible size dependency of $J_{max\ EQE}$ shift, which we have first demonstrated in this paper.

Regarding the consideration of surface recombination in the sidewall of μLEDs simulation, we have considered the surface recombination velocity in the sidewall, and the details are added

in the revised manuscript.

Modified Content

[Methods – Simulation – Line 542] The surface recombination velocity was set to 1.0×10^5 cm/s.

Fig. R6 Modified electron blocking layer. **a** Transmission electron microscope (TEM) images of QB 10.5 and QB 10.5 Balanced EBL. **b-f** simulation results of **b** energy band diagram **c** electron concentration **d** hole concentration **e** radiative recombination rate **f** electron current density of QB 10.5 and QB 10.5 Balanced EBL at the current density of 0.1 A/cm^2 .

[Reviewer’s Comment]

7. Fig. 4(b): why showing $J_S/J_B(V)$? While it is only $J_S(V)$ for Fig. 2(b).

[Response]

We appreciate reviewer #1 for pointing out this problem. This misunderstanding is due to our mistake made when presenting the same figure in Fig. 2c and Fig. 2f in the original submitted manuscript. We apologize for the confusion caused due to our mistake. Fig. R7 shows the J_S/J_B - V curve which can infer the ratio of surface current to bulk current. Fig. R7 shows a clear trend of J_S/J_B of QB 3.5 higher than QB 7.5, and QB 7.5 higher than QB 10.5 when the applied voltage is higher than 2.45V, implying that the dominance of surface current increases with increasing voltage as the QB thickness decreases.

Fig. R7 J_S/J_B as a function of voltage for three different samples.

Fig. R8 J_S as a function of voltage for QB 10.5 and QB 10.5 Balanced EBL

Additionally, to avoid confusion, the J_S - V curve and corresponding explanation of QB 10.5 and QB 10.5 Balanced EBL is also added to Fig. 4b (Fig. R8) in the revised manuscript.

Modified Content

Fig. R7 replaces Fig. 2c.

Fig. R8 and the corresponding explanation are added to Fig. 4b.

[Line 370] By analyzing the J - V characteristics using the same method mentioned in the previous section, we found that the J_S and J_S/J_B of QB 10.5 Balanced EBL at the forward bias smaller than 2.9 V (for J_S) and 2.95 V (for J_S/J_B) are lower than QB 10.5 as depicted in Fig. 4b and Fig. 4c, which indicates implies that the ratio of surface leakage current to the μ LEDs mesa sidewall is much lower smaller at small applied bias.

[Reviewer's Comment]

8. Fig. 4(e) not necessary.

[Response]

We value your comment on the necessity of Fig. 4e, which has no direct relationship with the content of the manuscript. As a response, we have deleted Fig. 4e in the original manuscript. The explanation related to this figure in the original manuscript is also deleted.

[Reviewer's Comment]

9. General comment on EQEmax values: The final current density at EQEmax obtained with the best structure (3 A/cm²) has a value equivalent as standard blue LED (using structure without pushing too much the droop to higher current density). Can you comment on this? Can you also comment on the fact that the EQEmax for all the structures does not change too much (between 2.5 and 4%)?

[Response]

[Redacted]

[Redacted]

[Redacted]

[Redacted]

[Redacted]

[Reviewer's Comment]

10. Device fabrication method: can you precise in the text that there is no passivation of the micro-LED sidewalls? Can you also precise the growth technique?

[Response]

We appreciate the reviewer for this helpful comment. As you point out, the description of the un-passivated sidewall surface may be not enough and needs to be described extensively. For this reason, we have added some phrases and sentences in the introduction and device fabrication method as below:

Modified Content

[Line 80] By utilizing these optimal structures, even without the sidewall passivation, the plot of resulting size-dependent EQEs showed a negligible $J_{max\ EQE}$ shift indicating a highly suppressed sidewall effect even in $10 \times 10 \mu\text{m}^2$ devices with the highest EQE in low current density (0.1 A/cm^2) ever reported.

[Methods – Device Fabrication – Line 499] For as-deposited devices, sidewall passivation of mesa such as KOH passivation is not conducted in order to observe the effect of epitaxial structure on device performance, and the default devices are as-deposited device unless mentioned.

Additionally, we have added the chemicals and process that is exposed to the sidewall during the fabrication of the devices in the device fabrication method:

[Methods – Device Fabrication – Line 508] Regarding the history of chemicals and fabrication processes exposed to the sidewall, the device sidewall is exposed to buffered oxide etch solution when removing the SiO_2 hard mask and is covered by Al_2O_3 through atomic layer deposition during the formation of isolation layer for p-type contact. The covered Al_2O_3 is then completely etched using a buffered oxide etch. During the metal pad lithography process, the sidewall is exposed to DNR-L300-40 which is a negative photoresist, and at last, acetone is exposed to the sidewall during the liftoff process.

Regarding the details of the growth technique, we have described the growth technique more in detail which could be found in the device fabrication methods in the manuscript and below:

[Methods – Epitaxy Growth – Line 471] All four InGaN-based epitaxialepitaxy structure (QB 3.5, QB 7.5, QB 10.5, and QB 10.5 Balanced EBL) were grown on 8-inch Si (111) wafer in a MOCVD chamber. A 200 nm high-temperature AlN buffer layer is grown. 20 pairs of AlGaN/AlN superlattice layers of thickness ranging from 150 nm to 170 nm are grown, followed by a 150 nm AlGaN buffer layer. AlGaN/AlN/GaN superlattice based buffer layers were grown on the substrate, followed by Then, an undoped GaN buffer layer with thickness of 1.5 μm , Si-doped n-type GaN with a thickness of 1.5 μm , and 20 pairs of $\text{In}_{0.02}\text{GaN/GaN}$

(1.5nm/2nm) superlattices for strain relief were grown. Then, 6 pairs of InGaN/GaN multiple quantum wells were grown according to their QB thickness specification. For the EBL structure, in the case of QB thickness-controlled samples (QB 3.5, QB 7.5, and QB 10.5), a 17 nm thick $\text{Al}_{0.2}\text{In}_{0.02}\text{Ga}_{0.78}\text{N}$ with $2 \times 10^{18} \text{ cm}^{-3}$ Mg p-type doping concentration was grown, and in case of QB 10.5 Balanced EBL, a 45 nm thick $\text{Al}_{0.12}\text{In}_{0.02}\text{Ga}_{0.86}\text{N}$ with $2 \times 10^{19} \text{ cm}^{-3}$ Mg p-type doping concentration was grown. At last, highly Mg-doped p-type layer and a very thin InGaN capping layer were grown.

[Reviewer's Comment]

11. Measurement method: there is no mention of the EDX measurements.

[Response]

Thank you for the kind reminder for the detailed measurement setup description. We have added the description of EDS measurement.

Modified Content

[Methods – Measurement method – Line 521] TEM images were obtained by JEM-2100F, and the EDS data were obtained by X-Max^N from Oxford Instruments.

Reviewer #2 (Remarks to the Author):

[Reviewer's Comment]

This manuscript reports the fabrication of μ LEDs with high emission efficiency and good thermal properties using an optimized design of QB thickness and EBL. The resulting μ LED devices exhibit a high EQE of 2.04% at 0.1 A/cm² for $10 \times 10 \mu\text{m}^2$ sizes. This result suggests that a new epitaxial approach can effectively reduce the μ LED pixel temperature, which can contribute to the production of high-resolution μ LED displays. The manuscript, however, includes unclear results and descriptions. Before the final decision of publication, the authors should address the following comments.

[Response]

We appreciate reviewer #2 for valuing our new epitaxial approach.

[Reviewer's Comment]

1. Wong et al, Appl. Phys. Express 12, 097004 (2019) and Sheen et al, Nature 608, 56 (2022) report much higher EQE for μ LEDs of $< 10 \times 10 \mu\text{m}^2$ size than that of this work. The authors should compare with the results of previous publications at fixed current density. “record high” word cannot be used without a clear analysis and comparison with the previous reports.

[Response]

We appreciate the comments on the comparison of EQE results from other works. The efficiency in the low-current density region is important in micro-display. The brightness of display devices is often in the range between $\sim 10^3$ to $\sim 10^4$ cdm⁻², where the current density for μ LEDs is normally operated between 0.02 A/cm² to 2 A/cm²¹¹. Not only the operation conditions, the self-heating issues, and the chromaticity shift in high current density region are also the strong reasons why the μ LEDs display should be operated in low current density. As the reviewer has requested a clear analysis and comparisons of the EQE, we compared the EQE results at particular current densities (0.1, 1, and 10 A/cm²) from various papers and summarized them in Table. R1. The EQE value of our device of QB 10.5 Balanced EBL shows the highest EQE at 0.1 A/cm² even though the maximum EQE is not high due to the low light extraction efficiency caused by Si (111) substrate, which means that our proposed epitaxial structure has higher potential in terms of EQE when the light extraction efficiency is enhanced and it is still very high even compared to other recent studies. Nevertheless, we have toned down to address the concern of reviewer #2, since the highest EQE at 0.1 A/cm² in our paper is lower than the maximum EQEs of other papers.

We also updated the benchmark showing our device performances after the sidewall passivation which is the results from your comment #2, and two more references of Wong et al.¹² and Sheen et al¹³. have been added as shown in Fig. R11.

Table. R1. **Summarizing table of various state-of-art devices and devices in this work.** The EQE is compared at the current density of 0.1, 1, 10 A/cm². Maximum EQE and $J_{max\ EQE}$ are also shown in the table. (Ley et al.⁸, Smith et al.⁹, Wong et al.¹², Olivier et al.¹⁰, Sheen et al.¹³)

Reference	Device Size (diameter or length)	Device Shape	Substrate	EQE at 0.1 A/cm ²	EQE at 1 A/cm ²	EQE at 10 A/cm ²	Maximum EQE	$J_{max\ EQE}$
Ley et al. (smallest device size)	2 μm	Circular	C-plane Sapphire	1.6%	8.2%	12.7%	13.3%	12 A/cm ²
Ley et al. (Similar size to our smallest device)	10 μm	Circular	C-plane Sapphire	0.25%	5.8%	9.2%	9.2%	13.8 A/cm ²
Smith et al. (smallest device size)	1 μm	Circular	Sapphire	N/A	0.6%	1.2%	2.4%	767 A/cm ²
Smith et al. (Similar size to our smallest device)	10 μm	Circular	Sapphire	0.6%	2.7%	5.7%	6.0 %	32.1 A/cm ²
Wong et al. (smallest device size)	10 μm	Rectangular	PSS	N/A	N/A	21.0%	23.7%	15.1 A/cm ²
Olivier et al. (smallest device size)	10 μm	Rectangular	C-plane Sapphire	N/A	0.3%	4.7%	5.2%	31.5 A/cm ²
Sheen et al. (smallest device size)	530 nm	Circular	Grown on c-plane sapphire but released	N/A	18.8%	21.0%	21.0%	10.2 A/cm ²
This work (asdep)	10 μm	Rectangular	Si (111)	2.08%	3.73%	3.93%	4.08%	4 A/cm²
This work (sidewall passivated)	10 μm	Rectangular	Si (111)	3.00%	4.19%	4.40%	4.45%	5 A/cm²

Fig. R11 Benchmark of the different epitaxy structures and state-of-the-art blue μLED devices. b-c benchmark of b EQE at $J = 0.1 \text{ A/cm}^2$ and c $J_{\text{max EQE}}$ for state-of-art blue μLED devices as a function of pitch size. (UCSB in blue⁹, UCSB in green⁸, UCSB in ocher¹², LETI¹⁰, Samsung Display¹³)

Modified Content

Table R2 is added to Supplementary Information, and Fig. R11 is updated to Fig. 5 in the main manuscript.

[Abstract – Line 25] As a result, the fabricated micro-light-emitting diodes μLEDs show a record-high external quantum efficiency EQE of 3.00% at 0.1 A/cm^2 for the pixel size of $10 \times 10 \mu\text{m}^2$ and a negligible size-dependency $J_{\text{max EQE}}$ shift during size reduction, which is extremely challenging due to the non-radiative recombination at the sidewall.

[Line 80] By utilizing these optimal structures, even without the sidewall passivation, the plot of resulting size-dependent EQEs showed a negligible $J_{\text{max EQE}}$ shift indicating a highly suppressed sidewall effect even in $10 \times 10 \mu\text{m}^2$ devices with the highest EQE in low current density (0.1 A/cm^2) ever reported.

[Line 409] It shows that μLED devices fabricated from QB 10.5 Balanced EBL show a record-high EQE of 2.04% (for as-dep) and 3.00% (for sidewall passivated) when the pixel size is $10 \times 10 \mu\text{m}^2$ at the low current density of 0.1 A/cm^2 and shows that the $J_{\text{max EQE}}$ is at a low current density with negligible size dependence, which is an important feature for ultra-high resolution μLED display.

[Line 461] As a result, we successfully fabricated μLEDs showing a record-high EQE of 3.00% and at 0.1 A/cm^2 for the pixel size of $10 \times 10 \mu\text{m}^2$ in GaN-based blue μLEDs and low $J_{\text{max EQE}}$ throughout the fabricated size with negligible size-dependency.

[Reviewer's Comment]

2. If the side wall passivation is properly used, it would benefit EQE. It is recommended that the authors add the data related to the change in the thermal and emission properties induced by the side wall passivation.

[Response]

We appreciate the constructive comments on the effect of sidewall passivation on the device performances in terms of thermal and emission properties, and we agree that the sidewall passivation will benefit the device performance of μ LEDs. As a response, we have conducted an additional experiment to observe the effect of sidewall passivation. The sidewall passivation scheme of devices is as below:

[Methods - Device Fabrication – Line 501] For the devices with the sidewall passivation, the μ LEDs were treated with KOH (2mol/L) at room temperature for 40 min after the ICP-RIE etching. After the sidewall passivation, the SiO₂ hard mask is removed and sequentially loaded into the ALD chamber for ALD passivation.

Fig. R12 shows the current density of different devices when the sidewall passivation is applied to the devices, and R13 shows the J_S and J_S/J_B curve for both as-deposited (asdep) and sidewall passivated devices. Comparing the J - V characteristics with Fig. S3 which shows the results from asdep devices, the gradual increase of current density was reduced dramatically as shown in Fig. R12e-f. Additionally, when comparing J_S and J_S/J_B for asdep and sidewall passivated devices shown in Fig. R13, a decrease in both J_S and J_S/J_B can be observed for sidewall passivated devices than asdep devices, which means a significant decrease in the surface current. To be mentioned, the trend of surface current in asdep devices (Fig. R13a-b) is still valid in the sidewall passivated devices, strongly convincing the validity of the surface current measurement method.

We also plotted the S parameter of the asdep and sidewall passivated device with a size of $10 \times 10 \mu\text{m}^2$ as shown in Fig. R14. From the figure, a decrease in the S parameter was observed especially in the low current density for all structures. However, while the S parameters for thin QB thickness have a more dramatic decrease such as QB 3.5, a smaller change occurred in QB 10.5 Balanced EBL, emphasizing that QB 10.5 Balanced EBL has more immunity to the sidewall efficiency degradation due to its epitaxial nature of lower sidewall current.

Fig. R15 shows the EQE-logarithmic current density curve for the sidewall passivated devices of QB 3.5, QB 7.5, QB 10.5, and QB 10.5 Balanced EBL. The results are summarized and compared with the devices without sidewall passivation in Fig. R16. From the figure, we observed enhancements both in maximum EQE and $J_{max\ EQE}$, which many studies have already reported. These results are due to the decrease of surface currents shown in Fig. R13 as the lateral diffusion of carriers imposing surface recombinations has substantially decreased. However, the increasing EQE trend of the epitaxial structure from QB 3.5 to QB 10.5 Balanced EBL is still valid even though the sidewall passivation is conducted, which further convinces that the optimization of epitaxial structure is not an option, but essentials for μ LEDs display.

Fig. R12 Logarithmic current density versus voltage characteristics of different sizes of μ LEDs with sidewall passivation. **a&e** QB 3.5. **b&f** QB 7.5. **c&g** QB 10.5. **d&h** QB 10.5 Balanced EBL.

Fig. R13 J_S and J_S/J_B curve as a function of voltage for the as-deposited (asdep) and sidewall passivated μ LEDs devices. **a** J_S -V curve **b** J_S/J_B -V curve for asdep devices. **c** J_S -V curve **d** J_S/J_B -V curve for sidewall passivated devices.

Fig. R14 S parameter for the as-deposited (asdep) and sidewall passivated μLEDs devices with the device size of $10 \times 10 \mu\text{m}^2$. **a** J_S -V curve **b** J_S/J_B -V curve for asdep devices. **c** J_S -V curve **d** J_S/J_B -V curve for sidewall passivated devices.

Fig. R15 EQE-logarithmic current density curve for sidewall passivated QB 3.5, QB 7.5, QB 10.5 and QB 10.5 Balanced EBL with pitch size from $80 \times 80 \mu\text{m}^2$ to $10 \times 10 \mu\text{m}^2$. c QB 3.5 d QB 7.5 e QB 10.5 f QB 10.5 Balanced EBL.

Fig. R16 Device performance comparison between the asdep and sidewall passivated device. a maximum EQE and **b** $J_{\max EQE}$ of asdep and sidewall passivated devices with different sizes.

As the reviewer has recommended, we have also measured the thermal profile after the sidewall passivation. Table. R2 shows the thermoreflectance microscopy image of QB 3.5 to QB 10.5 Balanced EBL both for devices with and without sidewall passivation with the different light output power (LOP) per area at different pitch sizes of $10 \mu\text{m}^2 \times 10 \mu\text{m}^2$ and $80 \mu\text{m}^2 \times 80 \mu\text{m}^2$. The resulting average temperature is summarized in Fig. R16. The resulting average temperature shows a trend that the sidewall passivated devices with lower average temperature than the devices without sidewall passivation both for $10 \mu\text{m}^2 \times 10 \mu\text{m}^2$ and $80 \mu\text{m}^2 \times 80 \mu\text{m}^2$ sized devices, especially at higher LOP per area of $1 \text{ W}/\text{cm}^2$. In the meanwhile, at a lower LOP per area of $0.1 \text{ W}/\text{cm}^2$, the average temperature has negligible change after the sidewall passivation, which emphasizes the importance of the necessity of low-current operation in μLEDs display.

Table. R2 Thermal profile of μ LEDs with a pitch size of $80 \times 80 \mu\text{m}^2$ and $10 \times 10 \mu\text{m}^2$ when the light output power per area is 0.1 W/cm^2 and 1 W/cm^2 for asdep and sidewall passivated devices of QB 3.5, QB 7.5, QB 10.5, and QB 10.5 Balanced EBL (BE) samples. The current density above the TRM images shows the corresponding current density. The scale bar is valid only within the μ LEDs mesa region since the thermoreflectance coefficient calibration was conducted for the mesa region.

Fig. R17 Average temperature of devices fabricated from QB 3.5, QB 7.5, QB 10.5, and QB 10.5 Balanced EBL (BE) with and without sidewall passivation at different sizes and LOP per area. The X corresponds to devices without passivation and O corresponds to devices with sidewall passivation. The pitch size presented in the figure showing 80 μm and 10 μm corresponds to 80 × 80 μm² and 10 × 10 μm², respectively.

Modified Content

Fig. R12-15, and R16a-b and Table. R2 is added to the Supplementary Information, and the corresponding explanation is also added.

[Line 281] To be mentioned, enhancements in device performances were observed after the sidewall passivation, but the trend of J_S/J_B and device performances of QB thickness-varied devices were still valid (see Supplementary Fig. 7-10 in the Supplementary Information), which means that the epitaxy engineering is not an option but a must for μLEDs display.

[Line 387] Furthermore, with the sidewall passivation, the resulting EQE has enhanced for QB 10.5 Balanced EBL in all device sizes while maintaining the negligible $J_{max\ EQE}$ shift (see Supplementary Fig. 7-10).

[In the Supplementary Information] In order to observe the impact of sidewall passivation on the surface current, we fabricated the devices with the sidewall passivation scheme. Supplementary Fig. 7 shows the J - V characteristics of differently sized μLEDs with sidewall passivation. Comparing the J - V characteristics with Supplementary Fig. 3 which shows the results from asdep devices, the gradual increase of current density was reduced dramatically as shown in Supplementary Fig. 7e-f. The J_S - V curve and J_S/J_B - V curve are shown in Supplementary Fig. 8 to compare the surface current. We observe a decrease in both J_S and J_S/J_B as shown in Supplementary Fig. 8a-b. It is worth mentioning that the trend of decrease of

J_S/J_B when the QB thickness increases is still valid even though the sidewall passivation has been conducted, thus the trend in QB 10.5 and QB 10.5 Balanced EBL. Supplementary Fig. 8c-f shows the S parameter of the asdep and sidewall passivated device with a size of $10 \times 10 \mu\text{m}^2$. From the figure, a decrease in the S parameter was observed especially in the low current density for all structures. However, while the S parameters for thin QB thickness have more a dramatic decrease such as QB 3.5, a smaller change occurred in QB 10.5 Balanced EBL, emphasizing that QB 10.5 Balanced EBL has more immunity to the sidewall efficiency degradation due to its epitaxy nature of less sidewall current.

[In the Supplementary Information] Supplementary Fig. 9 shows the EQE-logarithmic current density curve for the sidewall passivated devices of QB 3.5, QB 7.5, QB 10.5, and QB 10.5 Balanced EBL, and it is summarized and compared with the devices without sidewall passivation in Supplementary Fig. 10. From the figure, we observed enhancements both in maximum EQE and $J_{max\ EQE}$. This is because of the decrease in the surface current as shown in Supplementary Fig. 8. Furthermore, the increasing EQE trend of the epitaxial structure from QB 3.5 to QB 10.5 Balanced EBL is also still valid after the sidewall passivation, which further convinces that the optimization of epitaxial structure is not an option, but essentials for μLEDs display.

[In the Supplementary Information] Supplementary Table 2 shows the thermoreflectance microscopy image of QB 3.5 to QB 10.5 Balanced EBL both for devices with and without sidewall passivation with different light output power (LOP) per area at different pitch sizes of $10 \mu\text{m}^2 \times 10 \mu\text{m}^2$ and $80 \mu\text{m}^2 \times 80 \mu\text{m}^2$. The resulting average temperature is summarized in Supplementary Fig. 12. The resulting average temperature shows a trend that the sidewall passivated devices with lower average temperature than the devices without sidewall passivation both for $10 \mu\text{m}^2 \times 10 \mu\text{m}^2$ and $80 \mu\text{m}^2 \times 80 \mu\text{m}^2$ sized devices, especially at higher LOP per area of 1 W/cm^2 . In the meanwhile, at a lower LOP per area of 0.1 W/cm^2 , the average temperature has negligible change after the sidewall passivation, which emphasizes the importance of the necessity of low-current operation in μLEDs display.

[Reviewer's Comment]

3. The authors demonstrate $10\mu\text{m} \sim 80\mu\text{m}$ μLEDs . Recently, μLEDs with a size of less than $5 \mu\text{m}$ have been produced for high-resolution display. Therefore, it would be important to show thermal properties and EQE characteristic data for μLED with less than $5 \mu\text{m}$.

[Response]

Response: We appreciate the reviewer's comment on the minimum size of μLEDs demonstrated in the manuscript. To get to the point, we could not fabricate smaller-sized μLEDs due to the limitation of our photolithography equipment. We are aware of the recent works on μLEDs with a size of less than $5 \mu\text{m}$ and have tried to demonstrate a smaller size than $5 \mu\text{m}$. However, it was difficult to fabricate devices with a size smaller than $5 \mu\text{m}$ due to the alignment margin and depth of focus of our alignment system. For this reason, the metal lines are misaligned on the device sidewall, which makes the device electrically short as shown in Fig. R18. Even though the metal lines are well aligned on some occasions, the proportion of metal coverage on the mesa area is too high. As a result, under normal operation conditions, very less

light can be detected both in terms of human eyes and the integrated sphere. Additionally, the device structure for small-sized μ LEDs ($< 5 \mu\text{m}$) would be much more sensitive to the surrounding structures (metal, isolation etc.) in terms of temperature. For this reason, the temperature profile obtained for μ LEDs with a pitch size smaller than $5 \mu\text{m}$ would not be as significant in a real display situation because we are currently using the test device structure with a top electrode (which is the material with a high thermal conductivity) with a high fill factor as follows. Thus, our team is currently working on the InGaN/GaN-based μ LEDs display fabrication which is similar to our recent work in AlGaInP μ LEDs¹⁴ for temperature comparison in the real display case. Furthermore, our laboratory has ordered a photolithography alignment system which is capable of higher alignment precision, and we expect to fabricate μ LEDs devices with smaller pitch sizes.

Fig. R18 Optical microscope image of $5 \times 5 \mu\text{m}^2$. Due to the alignment margin of the lithography system in the laboratory, the metal line is mis-aligned and deposited on the sidewall. As a result, the device showed a shorted current-voltage when measuring the top and bottom contact.

Reviewer #3 (Remarks to the Author):

[Reviewer's Comment]

This article optimizes the performance of micro-LEDs at low current densities through epitaxial structure design and analyzes the underlying physical mechanisms. However, there has been a lot of research on the adjustment and analysis of the epitaxial structure of LED devices, and this article is not novel enough in this regard. What's more, the article has the following problems, which need to be further revised.

[Response]

Response: We sincerely appreciate reviewer #3 for raising sharp comments which are definitely essential for our manuscript. We are aware of numerous past pieces of research made on epitaxial structures of LEDs in various aspects including the quantum barriers and electron blocking layers, which we have mentioned in the introduction as follows: "Various epitaxial structure engineering methods including modification in quantum wells (QW)^{15,16}, quantum barriers (QB)^{17,18}, electron blocking layers (EBL)¹⁹⁻²¹, and growth substrates²²⁻²⁴ have been proposed to mitigate the efficiency droop in the high injection current density region." However, best to our knowledge, our submitted manuscript is the only demonstration work based on experimental data showing epitaxial structure strategies to overcome the size-dependent efficiency degradation issues in μ LEDs. Furthermore, while various post-growth optimizations such as sidewall passivation have been the mainstream of research in μ LEDs to enhance the efficiency and solve the size-dependent issues, we wanted to provide a new approach and to remind the μ LEDs academia that the epitaxial structure is still important as it was when solving the efficiency droop in the large-sized LEDs. Last but not least, the efficiency in the low-current density is extremely important in the μ LEDs display due to its normal operating condition between 0.02 A/cm^2 to 2 A/cm^2 ^{11,25}. Additionally, the self-heating issues and the chromaticity shift in high current density in the high current density region provide strong reasons for the operation of μ LEDs in low current density. Our proposed paper is the first paper in terms of experimental epitaxial design to enhance the device performance at the low current density. We hope you can reconsider the value of our work in these respects. Our detailed responses to your technical comments are provided below.

[Reviewer's Comment]

1. What is the selection principle of quantum barrier thickness in this work? As shown in the study, the performance of small-scale micro-LED is progressively improved as the thickness of the quantum barrier increases from 3.5 nm to 10.5 nm. But for thinner or thicker quantum barrier thicknesses, the author did not mention. Please clarify.

[Response]

We appreciate the reviewer for commenting on the selection principle of quantum barrier thickness. We have selected the quantum barrier thickness based on various literature from other groups which are mostly ranging from 2.6 nm to 16 nm. Among them, three quantum barrier thicknesses have been selected considering R_T , miniband formation and electric field in the quantum well as discussed before. Fig. R19 (from Fig.2e in the manuscript) shows the

graph summarizing the maximum EQE and the $J_{max\ EQE}$ of the QB 3.5, QB7.5, and QB 10.5. From the graph, we could see a trend of increase in maximum EQE when the QB thickness increases. For this reason, for a QB thickness lower than 3.5 nm, we expect a decrease in the maximum EQE and a higher shift of $J_{max\ EQE}$ when the device size is smaller. On the contrary, we can observe a saturating trend when the QB thickness increases from 7.5 nm to 10.5 nm in terms of maximum EQE and $J_{max\ EQE}$, which indicates R_T has been sufficiently suppressed at this thickness range. For this reason, we can expect an enhancement in EQE and $J_{max\ EQE}$ if the QB thickness is thicker than 10.5 nm, but would not be as dramatic when the QB size increased from 3.5 nm to 10.5 nm.

Fig. R19 $J_{max\ EQE}$ and maximum EQE of devices with different sizes.

To address the reviewer’s comment, we have mentioned the expectation of thinner and thicker thicknesses of QB.

Modified Content

[Line 290] From the experiment on QB thickness variation, it clearly suggests a severe size-dependent efficiency degradation happens with decrease of QB thickness, while the saturation of device performances could occur with increase of QB thickness.

[Reviewer’s Comment]

2. By optimizing the quantum barrier thickness and EBL structure, the authors obtained a relatively higher EQE at ultra-low current densities, which is very attractive for micro-display.

However, the peak EQE value of the blue micro-LED designed by this study is much lower than the reported researches [1], and even the $80\times 80\ \mu\text{m}^2$ device, the peak EQE is only $\sim 6\%$. What are the differences among them? In other words, are the optimization methods of this work applicable to different epitaxy designs?

[1] Wong, Matthew S., et al. "Size-independent peak efficiency of III-nitride micro-light-emitting-diodes using chemical treatment and sidewall passivation." *Applied Physics Express* 12.9 (2019): 097004.

[Response]

[Redacted]

[Redacted]

[Redacted]

[Reviewer's Comment]

3. In Fig. 1(f), it can be seen that the quantum barrier thickness has a huge effect on the PL wavelength. The authors should present EL spectra of different micro-LEDs at different current densities, which are important for actual display devices.

[Response]

We highly appreciate the comment regarding the EL spectra of different μ LEDs. As the

reviewer has commented, the EL spectra of μ LEDs are important. To address the reviewer's comment, we have added the summarized EL data in the main figures and the EL spectra in the Supplementary Information.

Fig. R21 Normalized electroluminescence summary spectra of devices fabricated from QB 3.5, QB 7.5, QB 10.5, and QB 10.5 Balanced EBL. The operating current density of the device is at **a** 0.1 A/cm², **b** 1 A/cm², and **c** 10 A/cm². **d-e** peak wavelength and FWHM of **d** QB3.5, QB 7.5, and QB 10.5. **e** QB10.5 and QB 10.5 Balanced EBL. The device sizes were 80 × 80 μ m².

Modified Content

Fig. R21**a-c** is added to the Supplementary Information and Fig.R21**c-d** is added to Fig. 2**f** and Fig. 4**e** of the main manuscript. The corresponding explanation is also added.

[Line 199] Fig. 2**f** depicts the electroluminescence (EL) peak wavelength and the full width at half maximum (FWHM) value as a function of current density for devices with different QB thicknesses (See supplementary Fig. 5 for EL spectra at different current densities). A smaller blue shift of EL peak was observed in QB 3.5 than QB 7.5, and QB 10.5, which is due to the difference in polarization effect as explained in Fig. 1**f**. Additionally, all three devices showed evidence of the screening effect and band-filling effect, but in varying degrees. Smaller screening effect and band-filling effect were observed in thinner QB thicknesses^{1,2}. Despite this factor, the peak wavelength shift and change in FWHM for QB 10.5 in the low current density region was negligible, which means no chromaticity change in low current density.

[Line 389] Fig. 4e shows the EL peak wavelength and FWHM value as a function of current density for devices with different QB thicknesses (See supplementary Fig. 5 for EL spectra at different current densities). The EL peak wavelength and FWHM of QB 10.5 Balanced EBL were similar to QB 10.5.

[In the Supplementary Information] Supplementary Fig. 6 shows the normalized electroluminescence (EL) spectra of devices fabricated from QB 3.5 to QB 10.5 Balanced EBL at different current densities. The EL wavelength trend is similar to the photoluminescence (PL) spectra, where QB 3.5 having a lower peak wavelength than QB 7.5, QB 7.5 lower than QB 10.5, and QB 10.5 similar to QB 10.5 Balanced EBL. A strong Fabry-Perot interference can be observed in the EL spectra due to the large difference in the refractive index of Si substrate, GaN and air²⁷. For this reason, we have extracted the peak wavelength and full width at half maximum (FWHM) in Fig. 2e and Fig. 4e after the Gaussian fitting of the EL spectra.

[Reviewer's Comment]

4. How do the authors consider the potential effect of quantum well thickness on EQE and $J_{\max EQE}$? Is there an optimized principle for thickness selection of both QW and QB?

[Response]

We appreciate the reviewer for this question. As the reviewer pointed out, QW thickness would be another important knob to achieve high EQE and low $J_{\max EQE}$, whereas the impact of this is very complicated. The thickness of QW in a wurtzite GaN-based crystal will provide a trade-off relationship between several important physical properties, resulting in a tuning difficulty than that of QB thicknesses.

First, from the zero-order Wentzel-Kramers-Brillouin approximation shown in equation (2), the tunneling rate of a carrier between QWs will decrease when the QW thickness increases. Additionally, thicker QWs have advantages in carrier confinement, where higher carrier density in the QW can be achieved at the same current densities²⁸. If these are the only cases, one can expect enhancements in the EQE and the $J_{\max EQE}$. It can apparently improve the electrical and optical characteristics of LEDs. However, this is not the only physical property in the epitaxy with a wurtzite crystal structure. What we should also consider is the Quantum Confined Stark Effect (QCSE).

$$R_T = \frac{\exp\left(-\int_0^{L_b} 2\hbar^{-1}\sqrt{2m^*\Delta u} dx\right)}{L_w} \sqrt{2E_0/m^*} \quad (2)$$

L_b : QB thickness, \hbar : reduced Planck's constant, m^* : effective carrier mass, Δu : QB energy height, L_w : QW thickness, and E_0 : ground energy state of QW

If the thickness of QW increases, the overlap of hole and electron wavefunctions decreases due to the QCSE. For this reason, the radiative recombination of the thicker QW can be decreased resulting in a decrease in the EQE. Additionally, the critical thickness of QW limits the QW thickness selection in GaN-based epitaxial structures. A thicker QW than 6 nm without lattice strain breaking is highly challenging for growing a high-quality blue InGaN/GaN LEDs epitaxy²⁹.

For this reason, an optimized balance of these components should be considered for a structure with high EQE, in which we think a thickness of approximately 3 nm is the optimized thickness, where this QW thickness is adopted in the majority of wurtzite InGaN/GaN LEDs. Therefore, in this work, we have extensively studied the impact of QB thicknesses. For the QB thicknesses, we believe that our optimizing principle is described in the manuscript.

Modified Content

To address the reviewer's comment, we have mentioned the QW thickness as follows:

[Line 293] For further optimization of device performances, QW thickness is one of the important parameters. However, QW thickness variation is more complicated because it also provides a trade-off between several important physical properties such as tunneling rate, barrier height, energy states, overlaps of carrier wavefunctions and critical thickness.

[Reviewer's Comment]

5. Some details need to be carefully checked and revised, such as Fig. 2(c) and (f) are exactly the same.

[Response]

We apologize for the mistake made. We have changed the Fig. 2c to J_S/J_B -V curve.

[Reviewer's Comment]

6. The authors should provide the average temperature data at different light output power densities (the corresponding current densities should be clearly given) to better study the difference between different epitaxial structures.

[Response]

We appreciate reviewer #3 for this comment. For the current densities of operation conditions, we have added the corresponding current densities above the TRM figures as shown in Table R3 and R4. For the average temperature at different light output power densities, we have added TRM images of QB 3.5 to QB 10.5 Balanced EBL at 0.1 W/cm^2 in Table R4. Additionally, we have also measured the average temperature of devices with sidewall passivation in Table R4. Fig. R22 summarizes the average temperature of devices in Table R4.

Table R3. Thermal profile of μ LEDs with a pitch size of $80 \times 80 \mu\text{m}^2$ and $10 \times 10 \mu\text{m}^2$ when the light output power per area is 1 W/cm^2 for QB 3.5, QB 7.5, QB 10.5, and QB 10.5 Balanced EBL (BE) samples. The current density above the TRM images shows the corresponding operation condition. The scale bar is valid only within the μ LEDs mesa region since the thermoreflectance coefficient calibration was conducted for the mesa region.

LOP per Area = 1 W/cm^2

Table. R4 Thermal profile of μ LEDs with a pitch size of $80 \times 80 \mu\text{m}^2$ and $10 \times 10 \mu\text{m}^2$ when the light output power per area is 0.1 W/cm^2 and 1 W/cm^2 for asdep and sidewall passivated devices of QB 3.5, QB 7.5, QB 10.5, and QB 10.5 Balanced EBL (BE) samples. The current density above the TRM images shows the corresponding current density. The scale bar is valid only within the μ LEDs mesa region since the thermoreflectance coefficient calibration was conducted for the mesa region.

Fig. R22 Average temperature of devices fabricated from QB 3.5, QB 7.5, QB 10.5, and QB 10.5 Balanced EBL (BE) with and without sidewall passivation at different sizes and LOP per area. The X corresponds to devices without passivation and O corresponds to devices with sidewall passivation. The pitch size presented in the figure showing 80 µm and 10 µm corresponds to 80 × 80 µm² and 10 × 10 µm², respectively.

Modified Content

Table R3 is updated to Table 1 in the manuscript. Table R4, Fig. R22 and corresponding explanations are added to the Supplementary Information.

[In the Supplementary Information] Supplementary Table 2 shows the thermoreflectance microscopy image of QB 3.5 to QB 10.5 Balanced EBL both for devices with and without sidewall passivation with different light output power (LOP) per area at different pitch sizes of 10 µm² × 10 µm² and 80 µm² × 80 µm². The resulted average temperature is summarized in Fig. S13. The resulting average temperature shows a trend that the sidewall passivated devices with lower average temperature than the devices without sidewall passivation both for 10 µm² × 10 µm² and 80 µm² × 80 µm² sized devices, especially at higher LOP per area of 1 W/cm². In the meanwhile, at a lower LOP per area of 0.1 W/cm², the average temperature has negligible change after the sidewall passivation, which emphasizes the importance of low-current operation in µLEDs display.

[Reviewer’s Comment]

7. In this paper, the junction current and sidewall current with different QB are analyzed in detail. However, for actual device fabrication, sidewall passivation has a great influence on the suppression of non-radiative recombination and should not be neglected. It is not known whether the sidewall passivation will affect the results of these analyses. Please clarify.

[Response]

We appreciate the comments on the influence on the surface current and the validity of proposed analysis method when surface passivation is conducted. In order to observe the impact of sidewall passivation on the surface current, we have conducted an additional experiment to observe the effect of sidewall passivation. The sidewall passivation scheme of devices is as below:

[Methods - Device Fabrication – Line 501] For the devices with the sidewall passivation, the μ LEDs were treated with KOH (2mol/L) at room temperature for 40 min after the ICP-RIE etching. After the sidewall passivation, the SiO₂ hard mask is removed and sequentially loaded into the ALD chamber for ALD passivation.

Fig. R23 shows the current density of different devices when the sidewall passivation is applied to the devices, and R24 shows the J_S and J_S/J_B curve for both as-deposited (asdep) and sidewall passivated devices. Comparing the J - V characteristics with Fig. S3 which shows the results from asdep devices, the gradual increase of current density was reduced dramatically as shown in Fig. R23e-f. Additionally, when comparing J_S and J_S/J_B for asdep and sidewall passivated devices shown in Fig. R24, a decrease in both J_S and J_S/J_B can be observed for sidewall passivated devices than asdep devices, which means a significant decrease in the surface current. To be mentioned, the trend of surface current in asdep devices (Fig. R24a-b) is still valid in the sidewall passivated devices, strongly convincing the validity of the surface current measurement method.

We also plotted the S parameter of the asdep and sidewall passivated device with a size of $10 \times 10 \mu\text{m}^2$ as shown in Fig. R25. From the figure, a decrease in the S parameter was observed especially in the low current density for all structures. However, while the S parameters for thin QB thickness have a more dramatic decrease such as QB 3.5, a smaller change occurred in QB 10.5 Balanced EBL, emphasizing that QB 10.5 Balanced EBL has more immunity to the sidewall efficiency degradation due to its epitaxy nature of lower sidewall current.

Fig. R23 Logarithmic current density versus voltage characteristics of different sizes of μ LEDs with sidewall passivation. **a&e** QB 3.5. **b&f** QB 7.5. **c&g** QB 10.5. **d&h** QB 10.5 Balanced EBL.

Fig. R24 J_S and J_S/J_B curve as a function of voltage for the as-deposited (asdep) and sidewall passivated μ LEDs devices. **a** J_S -V curve **b** J_S/J_B -V curve for asdep devices. **c** J_S -V curve **d** J_S/J_B -V curve for sidewall passivated devices.

Fig. R25 S parameter for the as-deposited (asdep) and sidewall passivated μ LEDs devices with the device size of $10 \times 10 \mu\text{m}^2$. **a** J_S -V curve **b** J_S/J_B -V curve for asdep devices. **c** J_S -V curve **d** J_S/J_B -V curve for sidewall passivated devices.

Modified Content

Fig. R23-25 and the corresponding explanation are added to the Supplementary Information.

[Line 281] To be mentioned, enhancements in device performances were observed after the sidewall passivation, but the trend of J_S/J_B and device performances of QB thickness-varied devices were still valid (see Supplementary Fig. 7-10 in the Supplementary Information), which means that the epitaxy engineering is not an option but a must for μ LEDs display.

[In the Supplementary Information] In order to observe the impact of sidewall passivation on the surface current, we fabricated the devices with the sidewall passivation scheme. Supplementary Fig. 7 shows the J -V characteristics of differently sized μ LEDs with sidewall passivation. Comparing the J -V characteristics with Supplementary Fig. 3 which shows the results from asdep devices, the gradual increase of current density was reduced dramatically as shown in Supplementary Fig. 7e-f. The J_S -V curve and J_S/J_B -V curve are shown in Supplementary Fig. 8 to compare the surface current. We observe a decrease in both J_S and J_S/J_B as shown in Supplementary Fig. 8a-b. It is worth mentioning that the trend of decrease of J_S/J_B when the QB thickness increases is still valid even though the sidewall passivation has

been conducted, thus the trend in QB 10.5 and QB 10.5 Balanced EBL. Supplementary Fig. 8c-f shows the S parameter of the asdep and sidewall passivated device with a size of $10 \times 10 \mu\text{m}^2$. From the figure, a decrease in the S parameter was observed especially in the low current density for all structures. However, while the S parameters for thin QB thickness have more a dramatic decrease such as QB 3.5, a smaller change was occurred in QB 10.5 Balanced EBL, emphasizing that QB 10.5 Balanced EBL has more immunity to the sidewall efficiency degradation due to its epitaxy nature of less sidewall current.

[Reviewer's Comment]

8. For the same QB thickness, $J_{\text{max EQE}}$ increases with the decrease in size. And for the same size, $J_{\text{max EQE}}$ decreases with the increase of QB thickness. More reasons need to be given.

[Response]

We thank you for this comment. We think this is due to the measurement interval. For all the device sizes, we measured the EQE in the same logarithmic increments of current. For example, when we measure the EQE between the current of 10 nA to 100 nA, there are 17 points between this range, so an increase of 5 nA is different between each interval in this range. However, in the next current range which is 100 nA to 1 μA and for the rest of current ranges, this logarithmic increase of current interval remains the same. Due to this reason, there are some gaps between the points and points in the EQE as shown in Fig. R26. When the gap is positioned near the maximum EQE, the accuracy of the $J_{\text{max EQE}}$ can be lower, but since we have to plot the benchmark based on the real measured value, there might be some ups and downs in $J_{\text{max EQE}}$ with sizes. However, we believe that this does not affect the trend of data.

Fig. R26 EQE versus logarithmic current density of QB 3.5 in different device sizes.

References

- [1] Lv, Q. et al. Realization of Highly Efficient InGaN Green LEDs with Sandwich-like Multiple Quantum Well Structure: Role of Enhanced Interwell Carrier Transport. *ACS Photonics* **6**, 130-138 (2019).
- [2] Yoo, Y.-S., Na, J.-H., Son, S. J. & Cho, Y.-H. Effective suppression of efficiency droop in GaN-based light-emitting diodes: role of significant reduction of carrier density and built-in field. *Scientific Reports* **6**, 34586 (2016).
- [3] Guan-Bo, L. et al. Effect of Quantum Barrier Thickness in the Multiple-Quantum-Well Active Region of GaInN/GaN Light-Emitting Diodes. *IEEE Photonics Journal* **5**, 1600207-1600207 (2013).
- [4] Nakamura, T. et al. Effect of Built-in Electric Field on Miniband Structure and Carrier Nonradiative Recombination in InGaAs/GaAsP Superlattice Investigated Using Photoreflectance and Photoluminescence Spectroscopy. *Energy Procedia* **102**, 121-125 (2016).
- [5] Chen, T.-C. & Bruce, R. Fundamentals of Laser Ablation of the Materials Used in Microfluidics. *Micromach. Tech. Fabr. Micro Nano Struct* (2012).
- [6] Geum, D.-M. et al. Strategy toward the fabrication of ultrahigh-resolution micro-LED displays by bonding-interface-engineered vertical stacking and surface passivation. *Nanoscale* **11**, 23139-23148 (2019).
- [7] Shim, J.-I. & Shin, D.-S. Measuring the internal quantum efficiency of light-emitting diodes: Towards accurate and reliable room-temperature characterization. *Nanophotonics* **7**, 1601-1615 (2018).
- [8] Ley, R. T. et al. Revealing the importance of light extraction efficiency in InGaN/GaN microLEDs via chemical treatment and dielectric passivation. *Applied Physics Letters* **116**, 251104 (2020).
- [9] Smith, J. M. et al. Comparison of size-dependent characteristics of blue and green InGaN microLEDs down to 1 μm in diameter. *Applied Physics Letters* **116**, 071102 (2020).
- [10] Olivier, F. et al. Influence of size-reduction on the performances of GaN-based micro-LEDs for display application. *Journal of luminescence* **191**, 112-116 (2017).
- [11] Jia, X. et al. A simulation study on the enhancement of the efficiency of GaN-based blue light-emitting diodes at low current density for micro-LED applications. *Materials Research Express* **6**, 105915 (2019).
- [12] Wong, M. S. et al. Size-independent peak efficiency of III-nitride micro-light-emitting-diodes using chemical treatment and sidewall passivation. *Applied Physics Express* **12**, 097004 (2019).
- [13] Sheen, M. et al. Highly efficient blue InGaN nanoscale light-emitting diodes. *Nature* **608**, 56-61 (2022).
- [14] Park, J., Geum, D.-M., Baek, W., Shieh, J. & Kim, S. Monolithic 3D sequential integration realizing 1600-PPI red micro-LED display on Si CMOS driver IC in 2022 *IEEE Symposium on VLSI Technology and Circuits (VLSI Technology and Circuits)*. 383-384 (IEEE, Year).
- [15] Lee, Y.-J., Chen, C.-H. & Lee, C.-J. Reduction in the efficiency-droop effect of InGaN green light-emitting diodes using gradual quantum wells. *IEEE Photonics Technology Letters* **22**, 1506-1508 (2010).
- [16] Maier, M., Köhler, K., Kunzer, M., Pletschen, W. & Wagner, J. Reduced nonthermal rollover of wide-well GaInN light-emitting diodes. *Applied Physics Letters* **94**, 041103 (2009).
- [17] Wang, C. H. et al. Hole transport improvement in InGaN/GaN light-emitting diodes by

- graded-composition multiple quantum barriers. *Applied Physics Letters* **99**, 171106 (2011).
- [18] Chang, J.-Y., Tsai, M.-C. & Kuo, Y.-K. Advantages of blue InGaN light-emitting diodes with AlGa_N barriers. *Optics letters* **35**, 1368-1370 (2010).
- [19] Yan Zhang, Y. & An Yin, Y. Performance enhancement of blue light-emitting diodes with a special designed AlGa_N/Ga_N superlattice electron-blocking layer. *Applied physics letters* **99**, 221103 (2011).
- [20] Wang, C. et al. Hole injection and efficiency droop improvement in InGa_N/Ga_N light-emitting diodes by band-engineered electron blocking layer. *Applied Physics Letters* **97**, 261103 (2010).
- [21] Kim, M.-H. et al. Origin of efficiency droop in Ga_N-based light-emitting diodes. *Applied Physics Letters* **91**, 183507 (2007).
- [22] Chiu, C.-H. et al. Reduction of efficiency droop in semipolar (1101) InGa_N/Ga_N light emitting diodes grown on patterned silicon substrates. *Applied physics express* **4**, 012105 (2010).
- [23] Bai, J. et al. Optical and polarization properties of nonpolar InGa_N-based light-emitting diodes grown on micro-rod templates. *Scientific Reports* **9**, 1-8 (2019).
- [24] Schmidt, M. C. et al. High power and high external efficiency m-plane InGa_N light emitting diodes. *Japanese journal of applied physics* **46**, L126 (2007).
- [25] Lu, S. et al. Designs of InGa_N Micro-LED Structure for Improving Quantum Efficiency at Low Current Density. *Nanoscale Research Letters* **16**, (2021).
- [26] Zhou, S., Cao, B., Liu, S. & Ding, H. Improved light extraction efficiency of Ga_N-based LEDs with patterned sapphire substrate and patterned ITO. *Optics & Laser Technology* **44**, 2302-2305 (2012).
- [27] Hums, C. et al. Fabry-Perot effects in In Ga_N/Ga_N heterostructures on Si-substrate. *Journal of Applied Physics* **101**, 033113 (2007).
- [28] Hajdel, M. et al. Dependence of InGa_N Quantum Well Thickness on the Nature of Optical Transitions in LEDs. *Materials* **15**, 237 (2021).
- [29] Holec, D., Costa, P., Kappers, M. & Humphreys, C. Critical thickness calculations for InGa_N/Ga_N. *Journal of Crystal Growth* **303**, 314-317 (2007).

We would hope that these revisions, described above, would meet the requirements that the reviewer would need for a publication in *Nature communications*.

Finally, we would appreciate the reviewer again for his/her valuable comments, which were indispensable in the appropriate revisions of the present paper.

Sincerely,

SangHyeon Kim,

Associate Professor, School of Electrical Engineering,

Korea Advanced Institute of Science and Technology (KAIST)

Daehak-ro 291, Yuseong-gu, Daejeon 34141, Korea

Phone: +82-42-350-7452

E-mail: shkim.ee@kaist.ac.kr

REVIEWERS' COMMENTS

Reviewer #1 (Remarks to the Author):

The authors have successfully answered to all my questions and comments. I thus recommend the publication of this manuscript in Nature Communications.

Reviewer #2 (Remarks to the Author):

The authors addressed previous concerns according to the reviewers' comments and improved the quality of the manuscript. Although the authors failed to fabricate 5 μ m microled pixel, this reviewer hopes to see the result through another manuscript in the near future.

Reviewer #3 (Remarks to the Author):

The authors have made significant effort to address the problems. The maximum EQE is not the highest, but the authors indeed have tried to push the advance of the micro-LED display. Such papers published in NC in micro-LED area will surely help the development of the InGaN micro-LED display area, and the hot micro-LED topic will help increase the impact of NC journal. I think the paper can be still accepted in the current version.